# Endothelin-1 signaling maintains glial progenitor proliferation in the postnatal subventricular zone

Katrina L. Adams [1], Giulia Riparini[1,2], Payal Banerjee [3], Marjolein Breur[4], Marianna Bugiani[5] & Vittorio Gallo [1✉]

Signaling molecules that regulate neurodevelopmental processes in the early postnatal subventricular zone (SVZ) are critical for proper brain development yet remain poorly characterized. Here, we report that Endothelin-1 (ET-1), a molecular component of the postnatal SVZ, promotes radial glial cell maintenance and proliferation in an autocrine manner via Notch signaling. Loss of ET-1 signaling increases neurogenesis and reduces oligodendrocyte progenitor cell proliferation (OPC) in the developing SVZ, thereby altering cellular output of the stem cell niche. We also show that ET-1 is required for increased neural stem cell and OPC proliferation in the adult mouse SVZ following demyelination. Lastly, high levels of ET-1 in the SVZ of patients with Cathepsin A-related arteriopathy with strokes and leukoencephalopathy correlate with an increased number of SVZ OPCs, suggesting ET-1's role as a regulator of glial progenitor proliferation may be conserved in humans. ET-1 signaling therefore presents a potential new therapeutic target for promoting SVZ-mediated cellular repair.

[1] Center for Neuroscience Research, Children's National Research Institute, Children's National Hospital, Washington, DC 20010, USA. [2] Department of Biomolecular Sciences, University of Urbino "Carlo Bo", Urbino, Italy. [3] Children's National Research Institute Bioinformatics Unit, Children's National Hospital, Washington, DC 20010, USA. [4] Department of Child Neurology, VU University Medical Center, Amsterdam, The Netherlands. [5] Department of Pathology, VU University Medical Center, Amsterdam, The Netherlands. ✉email: vgallo@childrensnational.org

Radial glial cells (RGCs), or neural stem cells (NSCs), give rise to a wide diversity of neuronal and glial cells during embryonic development. Following birth, neurogenesis and gliogenesis continue in specialized regions of the mammalian brain. One such area, the subventricular zone (SVZ) of the lateral ventricles (LVs), has been highly studied for its role in adult neurogenesis, particularly following injury and disease[1]. However, little attention has been paid to the postnatal SVZ during a critical period of neurodevelopment. During the weeks following birth, the highly proliferative embryonic ventricular zone transitions into the more quiescent adult SVZ niche[2]. RGCs detach their basal processes from the pial surface and transform into adult NSCs and ependymal cells, which together form the pinwheel structures of the adult SVZ[3–5]. During this period, a dorsal wave of oligodendrogenesis occurs, whereby oligodendrocyte progenitor cells (OPCs) migrate from the SVZ to the subcortical white matter (SCWM) and other brain regions[6]. This perinatal period of brain development is highly susceptible to injury, especially hypoxia-ischemia that often results from early preterm birth and congenital heart defects[7,8]. Therefore, increased understanding of the signaling molecules that regulate the developing postnatal SVZ is needed for insight into how these processes may be dysregulated under pathological conditions.

The small signaling peptide Endothelin-1 (ET-1) has been widely studied for its vasoconstrictor activity in endothelial cells, and recently been found to modulate development of neural crest cells[9], astrocytes[10], and Schwann cells[11]. ET-1 and the other Endothelin proteins (ET-2 and ET-3) bind two G-protein coupled receptors, Ednra and Ednrb, which are coupled to multiple second messenger systems. Although ET-1 expression was first reported in the central nervous system (CNS) three decades ago[12,13], very little investigation into its functional roles in CNS development has been performed. Our lab previously identified ET-1 as an inhibitor of OPC maturation in vitro[14] and in vivo following demyelination of SCWM in the adult rodent brain[15,16]. More recently, ET-1 was also found to regulate oligodendrocyte (OL) myelin sheath number in the adult mouse cortex and zebrafish[17]. Together, these studies indicate that ET-1 regulates multiple aspects of OL development and repair. However, the role of ET-1 within the SVZ stem cell niche remains unknown.

Here, we identify ET-1 as an essential regulator of progenitor proliferation and cell commitment in the postnatal SVZ. We find that RGCs produce ET-1, which acts in an autocrine fashion to promote RGC maintenance and proliferation through Notch signaling activation. Loss of ET-1 signaling increases postnatal neurogenesis of olfactory bulb (OB) interneurons from the SVZ. In addition, we show that RGC-derived ET-1 signals directly to OPCs within the SVZ to maintain their proliferation via upregulation of pro-progenitor factor Gsx1 and downregulation of OL maturation genes, S100b and Ust. We also find that ET-1 ablation reduces NSC and OPC proliferation in the adult mouse SVZ after focal demyelination of the SCWM, indicating that ET-1 regulates the SVZ regenerative response to injury. Finally, we show that elevated ET-1 protein in the adult human SVZ of Cathepsin A-related arteriopathy with strokes and leukoencephalopathy (CARASAL) patients correlates with increased numbers of astrocytes and OPCs, indicating that ET-1 signaling plays a similar role in the human SVZ.

## Results

**ET-1 is a molecular component of the SVZ stem cell niche.** While Endothelin proteins have been shown to regulate CNS regeneration[15,16], little is known regarding their expression and function during postnatal neurodevelopment. Using in situ hybridization (ISH), we analyzed the mRNA expression profile of the three Endothelin proteins and their two receptors (Ednra and Ednrb) over the first three weeks of postnatal mouse brain development. Both Edn1 and Ednrb were expressed highly in the SVZ, while Edn2, Edn3, and Ednra were not expressed at detectible levels (Supplementary Fig. 1a–t). These results were confirmed via RT-PCR using whole-brain and microdissected SVZ-derived cDNA (Supplementary Fig. 1u, v). To quantify expression levels, we performed qPCR for Edn1 and Ednrb mRNA in microdissected tissue from different regions of postnatal day 7 (P7) wild-type (WT) brains and found that both genes were expressed more than twofold higher in the SVZ and SCWM, compared with the cortex (Fig. 1a). Edn1 mRNA expression within the SVZ did not significantly change over the first month of postnatal life, while Ednrb expression within the SVZ significantly decreased between P9 and P18 (Fig. 1b, c). This suggests that ET-1 signaling is especially active during the first two postnatal weeks and that regulation of the Endothelin signaling pathway may occur at the receptor level.

To identify the cellular localization of ET-1 and Ednrb proteins within the SVZ, we performed immunohistochemistry (IHC) on WT P10 mouse brain tissue. RGCs, identified by immunostaining with anti-GFAP, -BLBP, and -Sox2 antibodies, expressed both ET-1 and Ednrb (Fig. 1d–k). Interestingly, a significant percentage of S100β+ cells lining the LVs also expressed ET-1 and Ednrb (Fig. 1g). These cells are likely a mix of immature ependymal cells and RGCs, as both cell populations express S100β at this developmental stage[18]. In contrast, ET-1 and Ednrb did not co-localize with the neuronal progenitor cell (NPC) marker doublecortin (Dcx) (Fig. 1l). However, Olig2+ NG2+ OPCs in the SVZ also expressed Ednrb (Fig. 1m), in agreement with previous reports[14–16]. Following quantification, we determined that the majority of RGCs in the dorsal postnatal SVZ expressed both ET-1 and Ednrb (Fig. 1f, g, j, k), whereas OPCs expressed only Ednrb (Fig. 1n). Overall, the Endothelin pathway is a molecular component of the postnatal SVZ stem cell niche and its cellular expression pattern suggests a role in glial progenitor development.

**Loss of ET-1 signaling reduces RGC proliferation.** To determine the role(s) of ET-1 signaling in the SVZ, we first sought to ablate expression of ET-1 in the developing postnatal SVZ. Using NestinCreER^T2; R26RYFP mice, we induced specific recombination within the SVZ following tamoxifen administration to P4 mouse pups. Interestingly, the highest levels of recombination occurred within the dorsolateral SVZ, therefore we restricted our analysis to this region for the majority of this study (Supplementary Fig. 2a, b). No recombination was observed within endothelial cells or SCWM astrocytes (Supplementary Fig. 2c). To ablate ET-1 expression, we crossed NestinCreER^T2 mice with ET-1 floxed mice that contain loxP sites flanking exon 2 (Fig. 2a). Analysis of P10 ET-1^flox/flox; NestinCreER^T2 mice (hereafter referred to as ET-1 cKO) confirmed significant knockdown of ET-1 protein within the SVZ (Supplementary Fig. 2d–f, n).

Since we identified RGCs as the primary producers of ET-1 protein within the developing postnatal SVZ (Fig. 1), we first investigated whether RGCs were affected following ET-1 ablation. Using whole mount SVZ preparations, we observed fewer VCAM1+ RGCs contacting the apical surface of the LVs in P11 ET-1 cKO animals, compared with WT controls (Fig. 2b, c). We confirmed this decrease by quantifying the number of BLBP+ Sox2+ RGCs in coronal sections of the dorsolateral SVZ of WT and ET-1 cKO mice (Fig. 2d, f). Interestingly, we observed no significant difference in the total number of S100β+ cells lining the LVs between WT and ET-1 cKO animals, suggesting that ependymal cells were not affected (Fig. 2e). We next investigated

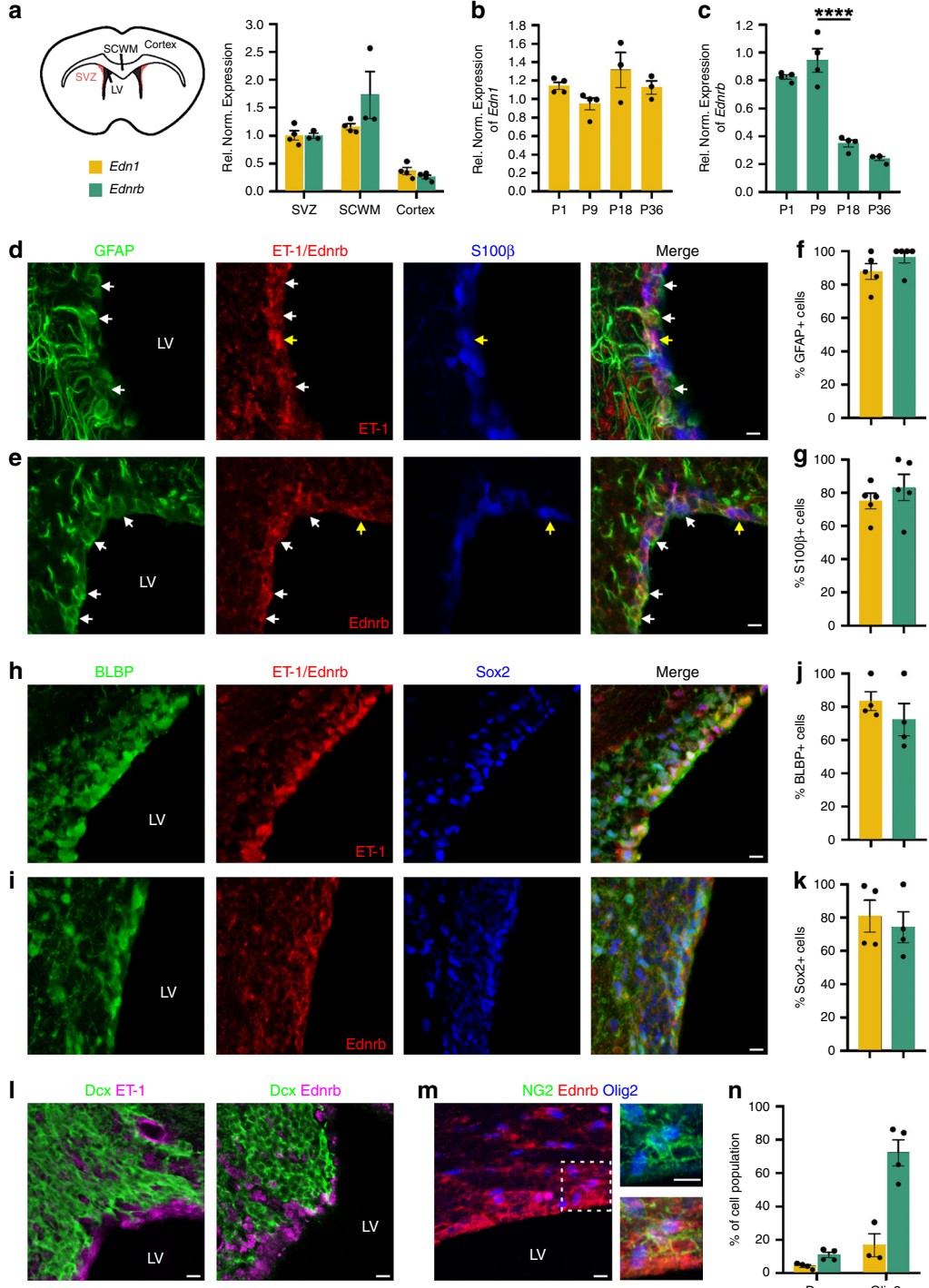

**Fig. 1 ET-1 and Ednrb are expressed in the developing postnatal SVZ. a** QPCR for *Edn1* and *Ednrb* mRNA levels in different regions of the postnatal day 7 (P7) mouse brain. SCWM and cortex values were compared with SVZ expression values, which were normalized to 1. (*Edn1*: $n = 4$ mice for each region, *Ednrb*: $n = 3$ mice for SVZ and SCWM, 4 mice for Cortex). QPCR for *Edn1* (**b**) and *Ednrb* (**c**) mRNA levels in the SVZ over the first postnatal month ($n = 4$ mice for each timepoint, except $n = 3$ for *Edn1* P18 and P36 timepoints). ****$p$ value < 0.0001 (one-way ANOVA with Tukey's multiple comparisons test). P1, P18, and P36 values were compared with P9 expression values, which were normalized to 1. Both ET-1 (**d**) and Ednrb (**e**) protein co-localize with RGC markers GFAP and S100β in the P10 mouse SVZ. White arrows point to ET-1+ or Ednrb+ GFAP+ cells. Yellow arrows point to ET-1+ or Ednrb+ S100β+ cells. Quantification of the percentage of GFAP+ cells (**f**) and S100β+ cells (**g**) in the dorsal WT SVZ at P10 that express ET-1 (yellow bar) or Ednrb (teal bar). ($n = 5$ WT mice for ET-1 and Ednrb). Both ET-1 (**h**) and Ednrb (**i**) protein co-localized with RGC markers BLBP and Sox2. Quantification of the percentage of BLBP+ cells (**j**) and Sox2+ cells (**k**) in the dorsal WT SVZ at P10 that express ET-1 or Ednrb. ($n = 4$ WT mice for ET-1 and Ednrb). **l** Neither ET-1 nor Ednrb is expressed by Dcx+ NPCs. **m** NG2+ Olig2+ OPCs in the SVZ and SCWM express Ednrb. **n** Quantification of the percentage of Dcx+ cells and Olig2+ cells in the dorsal SVZ that express ET-1 or Ednrb at P10. ($n = 4$ WT mice for ET-1 and Ednrb co-staining with Dcx, 3 WT mice for ET-1 co-staining with Olig2, and 4 WT mice for Ednrb co-staining with Olig2). All scale bars = 10 μm. LV lateral ventricle. Data are presented as mean values ± SEM. Source data are provided as a Source Data file.

whether this reduction in RGCs could be due to increased apoptosis and/or decreased cellular proliferation. We found no difference in the number of activated Caspase3+ cells in the SVZ at P10 in ET-1 cKO mice (Supplementary Fig. 3), indicating that loss of ET-1 signaling did not induce RGC apoptosis. To quantify cell proliferation, we administered BrdU to mice at P8 and P9 following tamoxifen injection at P4. We found that ET-1 cKO mice had significantly less BrdU+ BLBP+ cells in the SVZ at P10, compared with WT mice (Fig. 2d, g). Therefore, loss of ET-1 protein in the developing postnatal SVZ results in reduced RGC proliferation.

RGCs express the ET-1 receptor Ednrb, suggesting that ET-1 binds directly to Ednrb receptors on these cells to promote proliferation. To test this, we ablated Ednrb expression in RGCs by crossing *NestinCreER^T2; R26YFP* mice with *Ednrb^flox/flox* mice (hereafter referred to as Ednrb cKO). We confirmed knockdown of Ednrb protein within YFP+ cells and observed no compensatory change in Ednra protein expression (Supplementary Fig. 2g–m, o). Ednrb cKO mice recapitulated the ET-1 cKO RGC phenotype. At P11, Ednrb cKO mice also had reduced numbers of VCAM1+ RGCs in the anterior–dorsal SVZ, compared with WT mice (Fig. 2b, c). Furthermore, P10 Ednrb cKO mice had fewer BrdU+ BLBP+ cells in the dorsolateral SVZ, compared with WT mice (Fig. 2d, g). Overall, there were no significant differences between the ET-1 cKO and Ednrb cKO mice. Therefore, these results indicate that ET-1 signals directly to RGCs via the Ednrb receptor in an autocrine manner to maintain proliferation. Lastly, to determine whether loss of ET-1 signaling in RGCs at early postnatal ages has a lasting effect on the SVZ niche, we analyzed the dorsal SVZ of ET-1 cKO and Ednrb cKO mice at P28. Both ET-1 cKO and Ednrb cKO mice had significantly fewer VCAM1+ GFAP+ Sox2+ NSCs (or Type B cells) at P28, compared with WT controls (Fig. 2h, i). Therefore, ET-1 signaling in the developing postnatal SVZ is required for the correct density of NSCs in the adult SVZ niche. Interestingly, we also found that there was a reduction in the percentage of YFP+ (recombined) RGCs in the Ednrb cKO SVZ, compared with WT mice (70.33% of RGCs were YFP+ in WT mice versus 43.83% in Ednrb cKO mice). This indicates that the RGCs that escape deletion outcompete the poorly proliferating null cells.

**Loss of ET-1 signaling promotes postnatal neurogenesis.** RGCs within the developing postnatal SVZ generate both neuronal and glial progenitors that migrate throughout the brain and contribute to neural circuit plasticity. To determine whether decreased RGC proliferation following loss of ET-1 signaling affected cellular differentiation, we used the *R26RYFP* reporter to perform lineage tracing of RGCs in *NestinCreER^T2; R26RYFP* (WT), ET-1 cKO, and Ednrb cKO mice (Fig. 3a). At P10, we observed a significant decrease in the percentage of labeled cells that expressed the RGC marker BLBP and a corresponding increase in the percentage that expressed the NPC marker Dcx (Fig. 3b, c). This suggests that loss of autocrine ET-1 signaling promoted differentiation of RGCs into NPCs, thereby increasing postnatal neurogenesis. Interestingly, there was no significant change in the percentage of YFP-labeled cells that expressed Olig2 or S100β (Fig. 3b, c), indicating that Ednrb ablation in RGCs did not alter their differentiation into OPCs or ependymal cells, respectively. However, at P14 we found a significant decrease in the percentage of Olig2+ YFP+ cells in the Ednrb cKO mice (8.5 ± 1.4%), compared with WT mice (13.7 ± 0.7%) (*p* value = 0.0495; Welch's *t*-test). This suggests that reducing RGC number does impact SVZ oligodendrogenesis.

To determine whether a specific subtype of NPC was affected, we quantified Pax6+, Gsx2+, and Sp8+ NPC populations in the

dorsal SVZ and found that only the Sp8+ population was increased in both ET-1 cKO and Ednrb cKO mice, compared with WT controls (Fig. 3d, e). This may be because the *NestinCreER^T2* mouse strain largely induced recombination in the dorsal SVZ, where the majority of NPCs express Sp8. Importantly, we also observed an increase in SVZ-derived YFP+ NeuN+ neurons in the OB of both ET-1 cKO and Ednrb cKO mice at P28, compared with WT mice (Fig. 3f–h). We did not observe any YFP+ Caspase3+ cells within the OBs of WT, ET-1 cKO, and Ednrb cKO mice (Fig. 3i), suggesting that these excess neurons are stable. Therefore, ablation of ET-1 signaling in RGCs in the early postnatal SVZ enhances neurogenesis, thereby altering the cellular output of the stem cell niche and its ratio of glia to neurons.

**ET-1 promotes RGC maintenance via Notch signaling.** In order to better understand the mechanism by which ET-1 regulates SVZ cell commitment, we generated neurospheres from the SVZ of P8-P10 WT mice and treated them with 100 nM ET-1 (Fig. 4a). We isolated RNA 24 h after ET-1 treatment and performed RNAseq analysis. We identified 1160 differentially expressed genes (DEGs) between control and ET-1-treated neurospheres (Fig. 4b), using a threshold of an adjusted *p* value < 0.05. Analysis of these DEGs revealed that a large number of known RGC/NSC genes were upregulated following ET-1 treatment (*Nestin, Vcam1, Vimentin, Aldh1l1, Tnc, Fgfr2, Lif*) (Fig. 4c). This suggests that ET-1 maintains RGC identity by promoting expression of a "stem cell" gene network. Interestingly, we also observed that several proneural transcription factors were significantly downregulated following ET-1 treatment, including *Ascl1* and *Hes6* (Fig. 4c). Ascl1 has been widely characterized for its important role in neurogenesis[19]. Hes6 has been previously shown to promote neuronal differentiation[20,21] and inhibit cell proliferation[22]. *Sox8* was also significantly downregulated following ET-1 treatment and has been previously shown to regulate OL differentiation[23]. Together these results indicate that ET-1 promotes RGC identity both directly by upregulating stem cell genes and indirectly by decreasing cell differentiation genes.

To validate our RNAseq results, we performed a detailed analysis of WT SVZ neurospheres that were cultured in the presence or absence of ET-1 for 6 days (Supplementary Fig. 4a). ET-1 treatment significantly increased the average diameter of neurospheres formed by day 6 of culture (Supplementary Fig. 4b–d). We also observed an increase in the percentage of proliferating progenitor cells (Sox2+ Ki67+ cells) within the ET-1-treated neurospheres, compared with controls (Supplementary Fig. 4e–g). In agreement with our RNAseq results, we found that ET-1-treated neurospheres expressed higher levels of RGC proteins BLBP, Nestin, and VCAM1 (Supplementary Fig. 4h, i, l, m). In addition, ET-1 treatment reduced the percentage of Ascl1+ intermediate progenitor cells and Sp8+ Dcx+ NPCs (Supplementary Fig. 4j, k, n–q). Interestingly, we observed a slight increase in the percentage of Olig2+ PDGFRα+ OPCs within ET-1-treated neurospheres, compared with controls (Supplementary Fig. 4r), suggesting that ET-1 promotes glial progenitor fates over neuronal fates. Lastly, we treated the neurospheres with BrdU to label dividing cells and saw a significant increase in the percentage of BrdU+ cells after ET-1 treatment, compared with control neurospheres (Supplementary Fig. 4s). Therefore, ET-1 protein promotes RGC proliferation and blocks neuronal differentiation in vitro.

The Notch signaling pathway plays an important role in regulating RGC proliferation and differentiation in vivo[24,25]. We found that two components of the Notch pathway, *Jag1* and *Hey1*, were significantly upregulated following ET-1 treatment in our RNAseq dataset (Fig. 4d). Importantly, we observed

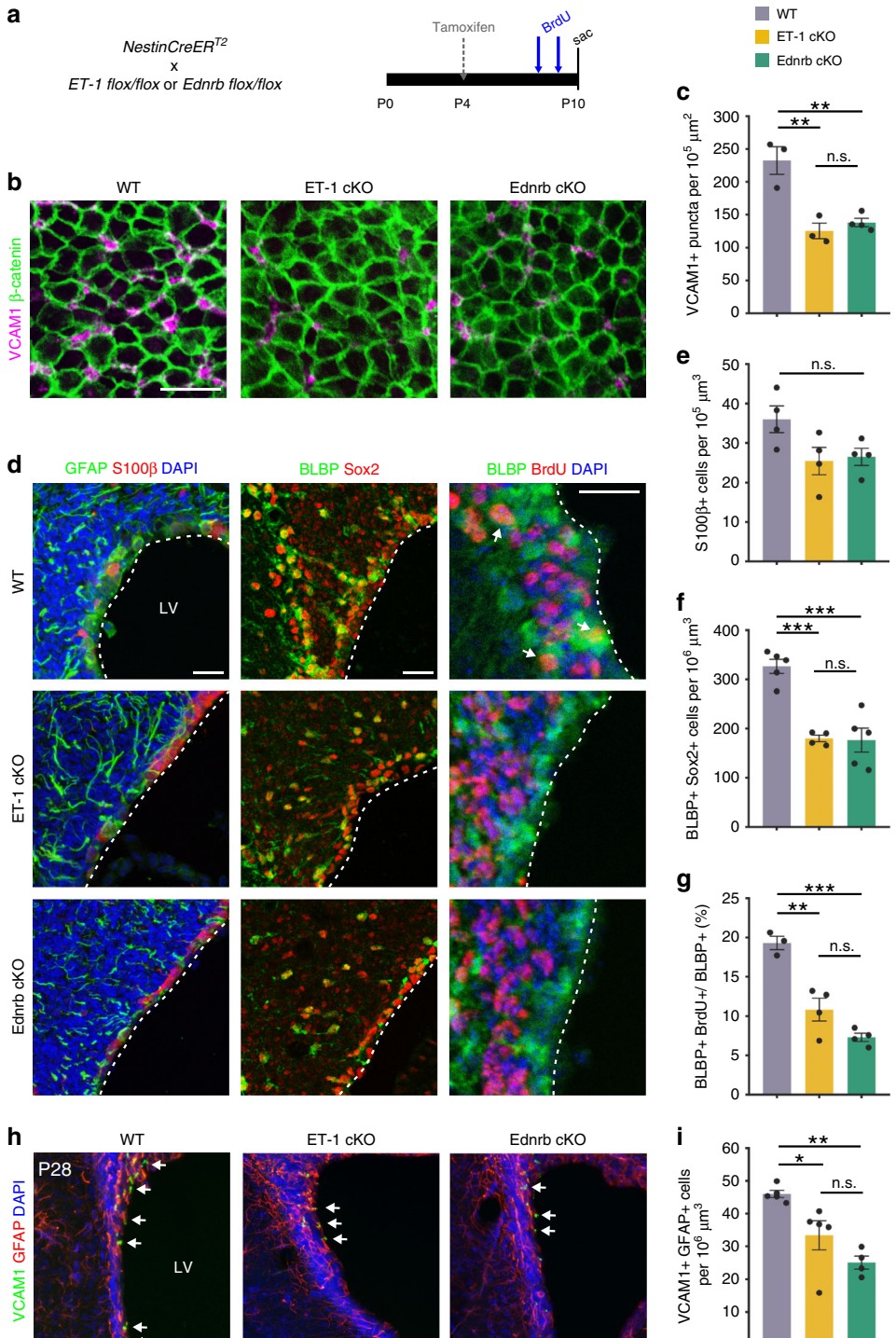

**Fig. 2 Ablation of ET-1 or Ednrb reduces radial glial number and proliferation. a** Strategy for conditional and inducible ablation of ET-1 or Ednrb in the postnatal SVZ. **b** Whole mount staining of RGC apical processes in P11 wildtype (WT), ET-1 cKO, and Ednrb cKO SVZ. **c** Quantification of the number of VCAM1+ RGCs contacting the apical surface of the SVZ. ($n = 3$ WT, 3 ET-1 cKO, 4 Ednrb cKO mice). **p value = 0.0022 (ET-1 cKO); **p value = 0.0031 (Ednrb cKO) (one-way ANOVA with Tukey's multiple comparisons test). **d** Coronal sections of P10 WT, ET-1 cKO, and Ednrb cKO SVZ. **e** Quantification of the number of S100β+ cells lining the lateral ventricles at P10. ($n = 4$ mice all groups). n.s. not significant (one-way ANOVA). **f** Quantification of the number of BLBP+ Sox2+ RGCs in the SVZ at P10. ($n = 5$ WT, 4 ET-1 cKO, 5 Ednrb cKO mice). ***p value = 0.0004 (WT versus ET-1 cKO); ***p value = 0.0002 (WT versus Ednrb cKO) (one-way ANOVA with Tukey's multiple comparisons test). **g** Quantification of the percentage of proliferating BLBP+ radial glia in the SVZ at P10. ($n = 3$ WT, 4 ET-1 cKO, 4 Ednrb cKO mice). **p value = 0.0014; ***p value = 0.0001 (one-way ANOVA with Tukey's multiple comparisons test). **h** Coronal sections of P28 WT, ET-1 cKO, and Ednrb cKO SVZ. Arrows point to VCAM1+ GFAP+ neural stem cells. **i** Quantification of the total number of VCAM1+ GFAP+ neural stem cells in the dorsal SVZ at P28. ($n = 5$ WT, 5 ET-1 cKO, 4 Ednrb cKO mice). *p value = 0.0266; **p value = 0.0015 (one-way ANOVA with Tukey's multiple comparisons test). All scale bars = 25 μm. LV lateral ventricle. Data are presented as mean values ± SEM. Source data are provided as a Source Data file.

significant upregulation of several Notch pathway genes (*Jag1*, *Notch1*, and *Hes5*) as early as 6 h post ET-1 treatment (Fig. 4e), suggesting that Notch signaling is an immediate downstream target of ET-1. To determine whether blocking ET-1 signaling in the SVZ reduces Notch signaling, we performed stereotaxic injections of the small molecule BQ788 (an Ednrb antagonist) into the LVs of P2 WT mice. At 6 h after BQ788 injection, we observed a significant reduction in *Jag1*, *Notch1*, and *Hes5* expression in the SVZ, compared with vehicle controls (Fig. 4f). This led to a decrease in total number of RGCs and OPCs, and an increase in NPCs, within the dorsolateral SVZ at 48 h post BQ788 injection (Supplementary Fig. 5), recapitulating the ET-1 cKO and Ednrb cKO phenotypes. Lastly, we also observed significant reductions in *Jag1*, *Notch1*, and *Hes5* expression in the SVZ of ET-1 cKO mice at P7 following tamoxifen injections (Fig. 4g), indicating that reduced Notch signaling underlies the decrease in RGC number and proliferation seen in these animals. Therefore, RGC-derived ET-1 acts in an autocrine manner to activate Notch signaling, thereby maintaining stem cell identity and proliferation.

**ET-1 directly promotes OPC proliferation in the SVZ**. In addition to RGCs, OPCs within the SVZ also express the Ednrb receptor (Fig. 1m). To determine whether ET-1 signaling also regulates OPC development in vivo, we analyzed OPCs within the SVZ of ET-1 cKO mice (Fig. 5a). Interestingly, ET-1 cKO mice had significantly less Olig2+ NG2+ OPCs in the dorsolateral SVZ at P10, compared with WT mice (Fig. 5b, c). Furthermore, both the total number of proliferating OPCs (Olig2+ Ki67+ cells) and the percentage of proliferating OPCs (Olig2+ Ki67+/Olig2+ cells) were significantly reduced in ET-1 cKO mice, compared with WT mice (Fig. 5b–d). Thus, ET-1 ablation reduces OPC proliferation in the developing postnatal SVZ.

To determine whether ET-1 signals directly to the Ednrb receptor on OPCs, we ablated receptor expression by crossing *PDGFRαCreER^T2; R26RYFP* mice with *Ednrb^flox/flox* mice (hereafter referred to as Ednrb OPC-cKO) (Fig. 5e). We confirmed that Ednrb protein was decreased in YFP+ OPCs (Supplementary Fig. 6). We found that Ednrb OPC-cKO mice had reduced numbers of Olig2+ NG2+ OPCs in the dorsolateral SVZ at P10, compared with WT mice (Fig. 5f, g). Importantly, the percentages of NG2+ and Ki67+ YFP+ OPCs were also reduced in the SVZ of Ednrb OPC-cKO mice, compared with WT mice (Fig. 5f, h). Therefore, ET-1 directly binds to Ednrb receptors on OPCs to promote their proliferation and maintain their progenitor state within the developing postnatal SVZ.

We then asked whether ET-1 overexpression promotes OPC proliferation in the SVZ. Using organotypic brain slice cultures from WT mice, we found that the addition of exogenous ET-1 resulted in an increase in Olig2+ NG2+ OPCs within the SVZ, compared with control slices (Fig. 5i–k). Furthermore, ET-1 treatment increased the total number of BrdU+ NG2+ Olig2+ cells within the SVZ (Fig. 5j, l), as well as the percentage of proliferating OPCs (Fig. 5m). Therefore, ET-1 overexpression increases the pool of proliferating OPCs within the SVZ.

**ET-1 induces downstream changes in OPC gene expression**. The effect of ET-1 on OPC proliferation has not been previously described and therefore prompts the question—what are the downstream effectors of ET-1 signaling in OPCs? Are these effectors the same or different from those in RGCs? Previous work in our lab found that exogenous ET-1 promoted ERK, P38MAPK, and CREB phosphorylation in cortical rat OPC cultures[14]. These are major signaling pathways that regulate a multitude of cellular behaviors, including growth, proliferation,

and differentiation[26]. To determine specific downstream molecular targets of ET-1 signaling in OPCs, we generated cultures of SVZ OPCs from P7–P8 WT mice via immunopanning for NG2+ cells and treated them with 100 nM ET-1 (Fig. 6a). Since ET-1 upregulated Notch signaling components in RGCs, we first analyzed *Jag1*, *Notch1*, and *Hes5* mRNA expression at 24 h post ET-1 treatment but found no significant changes (Supplementary Fig. 7). We then performed RNAseq on control and ET-1-treated OPC cultures to generate an unbiased list of ET-1 regulated genes. Following removal of low expression genes and applying a threshold (adjusted *p* value < 0.1), we identified 72 DEGs between control and ET-1-treated OPCs (Fig. 6b). Upstream regulator analysis of these DEGs using Ingenuity Pathway Analysis (IPA) predicted several signaling pathways, including ET-1, CREB1, and ERK (Fig. 6c), which is consistent with our previous findings. We then compared the OPC DEGs with our neurosphere DEGs to identify shared and distinct molecular pathways. Interestingly, half of the DEGs in our OPC dataset were also present in our neurosphere dataset (Fig. 6d). This included the stem/progenitor gene *Nestin*, as well as many other genes with unknown roles in OPC development (Fig. 6e).

To identify OPC-specific mechanisms of ET-1 signaling, we focused on the 41 DEGs changed only within our OPC dataset (Fig. 6f) and screened the genes for expression within the OL lineage using the Brain RNAseq database[27]. This revealed several interesting candidate genes, including *Ust* and *S100b*, which were downregulated in ET-1-treated OPC cultures and show higher expression in newly formed OLs, compared with OPCs. We also identified several genes that were upregulated in ET-1-treated OPC cultures that exhibit higher expression in OPCs, compared with OLs—including the transcription factor *Gsx1*. To validate our RNAseq results and determine whether these genes are regulated by ET-1 signaling in vivo, we performed RNAscope for *S100b*, *Ust*, and *Gsx1* expression within SVZ OPCs in WT and Ednrb OPC-cKO mice. We found a significant increase in the percentage of YFP+ OPCs that expressed *S100b* and *Ust* in Ednrb OPC-cKO mice, compared with WT mice (Fig. 6g–h). Furthermore, we observed a significant decrease in the percentage of *Gsx1*+ OPCs in the SVZ of Ednrb OPC-cKO mice, compared with WT mice (Fig. 6g–h). Therefore, ET-1 signaling regulates the expression of these genes within SVZ OPCs via signaling to the Ednrb receptor. While the function of *Ust* remains uncharacterized in OLs, *S100b* has been previously reported to promote OL maturation[28]. Conversely, one study reported that *Gsx1* mutant OPCs exhibited reduced proliferation, suggesting that it promotes OPC proliferation[29]. Therefore, ET-1 likely both upregulates progenitor factors and downregulates differentiation factors in OPCs, in a manner reminiscent of its activity in RGCs.

**ET-1 regulates glial development in the SCWM**. Interestingly, *Edn1* and *Ednrb* mRNA levels within the SCWM at P7 are similar to those of the SVZ (Fig. 1a), and both GFAP+ astrocytes and Olig2+ OL lineage cells expressed ET-1 protein in the corpus callosum (CC) at P10 (Supplementary Fig. 8a). This suggests that ET-1 regulates glial development and/or maturation in the SCWM, in addition to its role in the SVZ. To determine whether ET-1 signaling regulates OPC maturation in the CC, we analyzed Ednrb OPC-cKO mice at P14 (Supplementary Fig. 8b). We found a significant decrease in NG2+ YFP+ OPCs in the CC, along with an increase in CC1+ YFP+ mature OLs (Supplementary Fig. 8c, d). This corresponded with a significant increase in MBP expression within the CC of Ednrb OPC-cKO P14 mice, compared with WT controls (Supplementary Fig. 8c, d). Therefore, blocking ET-1 signaling via ablation of the Ednrb receptor on OPCs promotes OL maturation in the developing CC.

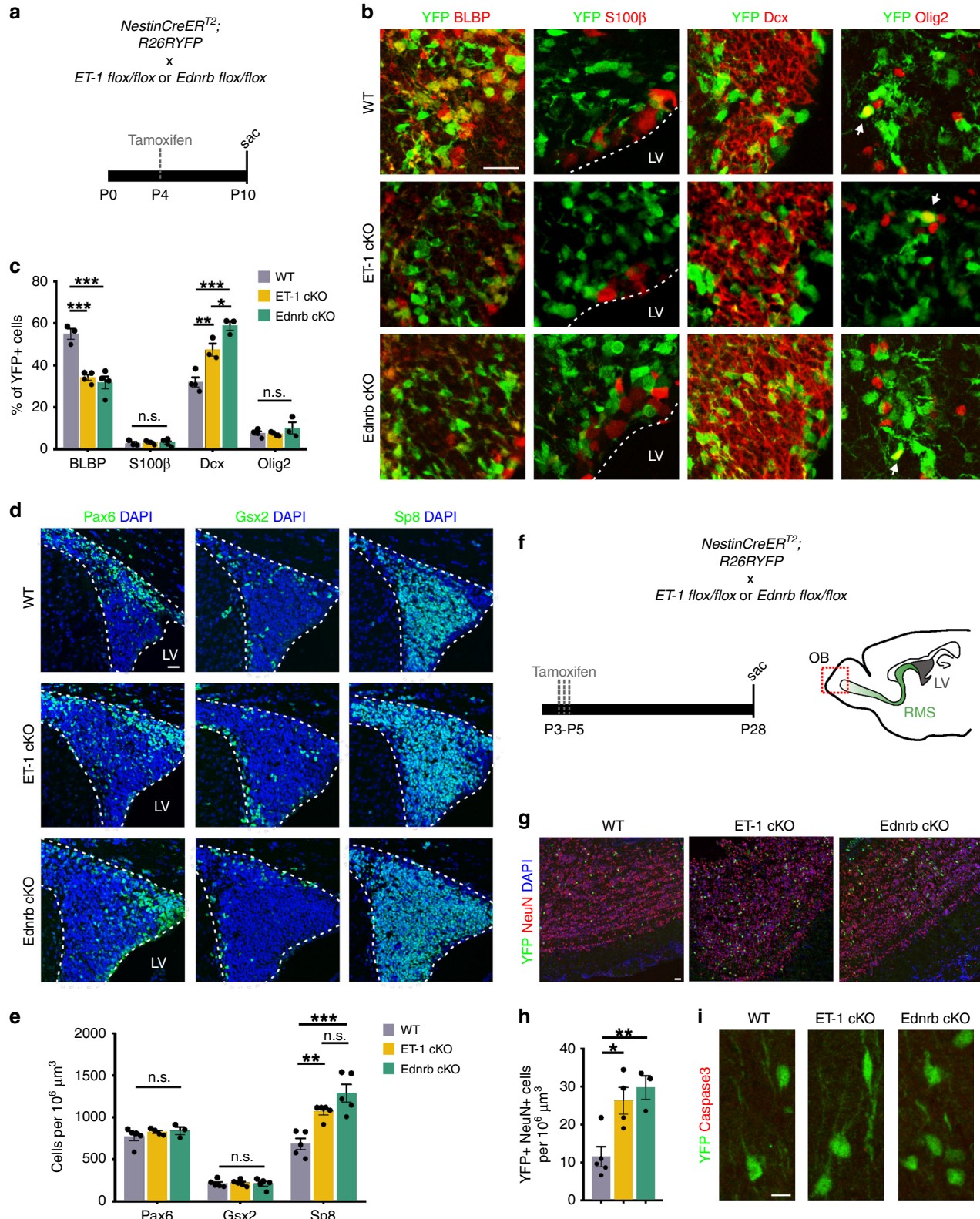

We next asked whether ET-1 regulates astrocyte development within the SCWM. Using an astrocyte specific cre-driver (*GFAPCreER[T2]*) crossed to our ET-1 floxed mice, we ablated ET-1 from astrocytes in the SCWM (Supplementary Fig. 9a). Interestingly, we found no changes in the percentage of Sox9+ astrocytes that expressed GFAP or Aldh1L1 (Supplementary Fig. 9b–d), indicating that ET-1 does not regulate astrocyte

maturation in the SCWM at early postnatal ages. However, we did observe a significant decrease in the percentage of proliferating (Ki67+ Sox9+) astrocytes following ET-1 ablation (Supplementary Fig. 9b, e). This was only observed in the cingulum region of the SCWM, as there was no change in astrocyte proliferation in the CC (Supplementary Fig. 9e). Therefore, ablation of ET-1 signaling reduces astrocyte proliferation in specific regions of the SCWM.

**Fig. 3 Ablation of ET-1 or Ednrb in the SVZ increases neurogenesis. a** Experimental strategy. **b** Coronal sections of YFP+ cells in WT, ET-1 cKO, and Ednrb cKO SVZ at P10. Scale bars = 25 μm. **c** Quantification of the YFP+ cell lineage in the SVZ at P10 (BLBP: $n = 3$ WT, 4 ET-1 cKO, 4 Ednrb cKO mice. S100β: $n = 3$ WT, 3 ET-1 cKO, 4 Ednrb cKO mice. Dcx: $n = 4$ WT, 3 ET-1 cKO, 3 Ednrb cKO mice. Olig2: $n = 4$ WT, 4 ET-1 cKO, 3 Ednrb cKO mice.). ***$p$ value = 0.0009 (BLBP expression between WT and ET-1 cKO); ***$p$ value = 0.0004 (BLBP expression between WT and Ednrb cKO); **$p$ value = 0.0072 (Dcx expression between WT and ET-1 cKO); ***$p$ value = 0.003 (Dcx expression between WT and Ednrb cKO); *$p$ value = 0.0390 (Dcx expression between ET-1 cKO and Ednrb cKO), n.s. not significant (one-way ANOVA with Tukey's multiple comparisons test). **d** Coronal sections of WT, ET-1 cKO, and Ednrb cKO SVZ at P10, staining for different NPC populations. Scale bars = 25 μm. **e** Quantification of NPC populations in the dorsal SVZ at P10 (Pax6: $n = 5$ WT, 4 ET-1 cKO, 3 Ednrb cKO mice. Gsx2: $n = 5$ mice all groups. Sp8: $n = 5$ mice all groups.). **$p$ value = 0.0090; ***$p$ value = 0.0003 (one-way ANOVA with Tukey's multiple comparisons test). **f** Experimental strategy. **g** Sagittal sections of YFP+ cells in olfactory bulb (OB) of WT, ET-1 cKO, and Ednrb cKO mice at P28. Scale bars = 25 μm. **h** Quantification of total YFP+ NeuN+ cells per area in OB ($n = 5$ WT, 4 ET-1 cKO, and 3 Ednrb cKO mice). *$p$ value = 0.0163; **$p$ value = 0.0079 (one-way ANOVA with Tukey's multiple comparisons test). **i** Magnified images of YFP+ neurons in OB of WT, ET-1 cKO, and Ednb cKO mice that are negative for Caspase3 staining. Images are representative of 5 WT, 4 ET-1 cKO, and 4 Ednrb cKO mice. Scale bar = 10 μm. LV lateral ventricle. Data are presented as mean values ± SEM. Source data are provided as a Source Data file.

**ET-1 signaling is reactivated in the adult SVZ after injury.** Developmental signaling pathways are often reutilized in the adult brain during regeneration following injury or disease. Demyelination of the SCWM in the adult rodent brain has been shown to result in increased oligodendrogenesis from the SVZ[30–32], as well as reactivation of critical OL developmental signals such as EGF[33]. Therefore, we asked if ET-1 regulates proliferation in the adult SVZ following focal demyelination of the SCWM. We first analyzed adult WT mice at 7 days post lysolecithin (LPC)-induced demyelination and found increased levels of ET-1 protein in the SVZ, compared with contralateral saline controls (Fig. 7a). The ET-1 protein largely co-localized with markers of NSCs (GFAP+ Sox2+ cells) (Fig. 7a). NSCs also expressed Ednrb after both saline and LPC injections, and we did not observe a significant change in Ednrb expression level following demyelination (Fig. 7a). We then performed focal demyelination of the SCWM in WT and ET-1 cKO mice and analyzed SVZ cell populations 7 days after injection (Fig. 7b). We observed an increase in the number of proliferating NSCs (Sox2+ GFAP+ Ki67+ cells) in the SVZ of WT animals following LPC injection, compared with saline controls (Fig. 7c, d). Interestingly, this increase was lost in the ET-1 cKO mice, suggesting that ET-1 plays a predominant role in promoting NSC proliferation in the SVZ following demyelination (Fig. 7c, d). Furthermore, ET-1 cKO mice displayed a significant decrease in the number of proliferating OPCs within the SVZ following LPC injection, in contrast to WT mice (Fig. 7c, e). This effect appears to be specific to the SVZ, as ablation of ET-1 from SCWM astrocytes did not alter OPC proliferation within the white matter lesion itself at 7 days post LPC injection (Supplementary Fig. 10). Lastly, there were no significant changes in the number of Sp8+ Dcx+ NPCs within the dorsal SVZ between WT and ET-1 cKO mice, both after saline and LPC injections (Fig. 7c, f). Therefore, these results indicate that ET-1 signaling is reactivated in the adult SVZ following SCWM demyelination to promote both NSC and OPC proliferation during remyelination, without altering neurogenesis, at least within this time window.

To determine whether ET-1 activates the same downstream pathways in the adult SVZ after demyelination, we performed RNAscope for several genes identified from our developmental experiments. We found that ET-1 cKO mice had reduced *Jag1* and *Hes5* expression in the dorsal SVZ following LPC injection, compared with WT mice (Fig. 7g–i). Interestingly, OPCs within the dorsal SVZ of WT mice greatly increased their expression of *Gsx1* following LPC injection, and this increase was lost in ET-1 cKO mice (Fig. 7g, j). Lastly, we observed an increase in the percentage of *S100b*+ OPCs in the SVZ of ET-1 cKO mice following LPC injection, compared with WT mice (Fig. 7g, k). Together these results demonstrate that developmental ET-1

pathways are reactivated in the adult SVZ following demyelination to promote NSC and OPC proliferation.

**ET-1 signaling is conserved in the human SVZ.** While ET-1 expression has been previously reported throughout different regions of the human brain[13], the expression and function of ET-1 in the human SVZ remains uncharacterized. In CARASAL, an adult-onset leukoencephalopathy, patients display ischemic and hemorrhagic strokes with slow cognitive decline and extensive white matter damage in the brain[34]. Using exome sequencing, Bugiani et al. identified one disease variant in the *CTSA* gene, which encodes for the enzyme Cathepsin A[34]. One function of Cathepsin A is to degrade ET-1, and CARASAL patients displayed increased levels of ET-1 in white matter astrocytes[34]. We wondered whether ET-1 was also increased in the SVZ of these patients, and if so—whether there would be any cellular effects. We found an overall increase in ET-1 levels in the SVZ of CARASAL patients (Fig. 8a–c). SVZ astrocytes appeared to be the primary producers of ET-1 protein as we observed high co-localization between GFAP and ET-1 (Fig. 8d, e). We also observed a significant increase in the number of GFAP+ astrocytes in the SVZ of CARASAL patients compared with age-matched controls (Fig. 8f). Interestingly, and consistent with our findings in the developing mouse SVZ, the number of PDGFRα+ OPCs in the SVZ of CARASAL patients was also increased. (Fig. 8g–i). Therefore, we conclude that ET-1 is a molecular component of the human SVZ and its functional role is likely conserved between the mouse and human postnatal SVZ.

**Summary.** Taken together, these results predict a model of ET-1 signaling in the developing postnatal SVZ that includes (1) autocrine ET-1 signaling within RGCs that activates Notch signaling, resulting in RGC maintenance and proliferation and (2) paracrine ET-1 signaling from RGCs to OPCs in the SVZ, resulting in upregulation of progenitor/proliferation genes and downregulation of OL maturation genes (Fig. 9). These pathways appear to be reactivated in the adult mouse SVZ following demyelination and are likely conserved within the human SVZ as well.

## Discussion

The postnatal SVZ continues to generate new neuronal and glial progenitor cells that migrate to different regions of the brain, including OPCs that migrate from the dorsal SVZ to the developing SCWM. Signaling molecules that regulate these processes during the perinatal period remain largely uncharacterized. We found that the signaling peptide ET-1 regulates glial progenitor populations in both the developing postnatal SVZ and the adult SVZ following injury, promoting both RGC and OPC proliferation and preventing their differentiation. Interestingly, we show

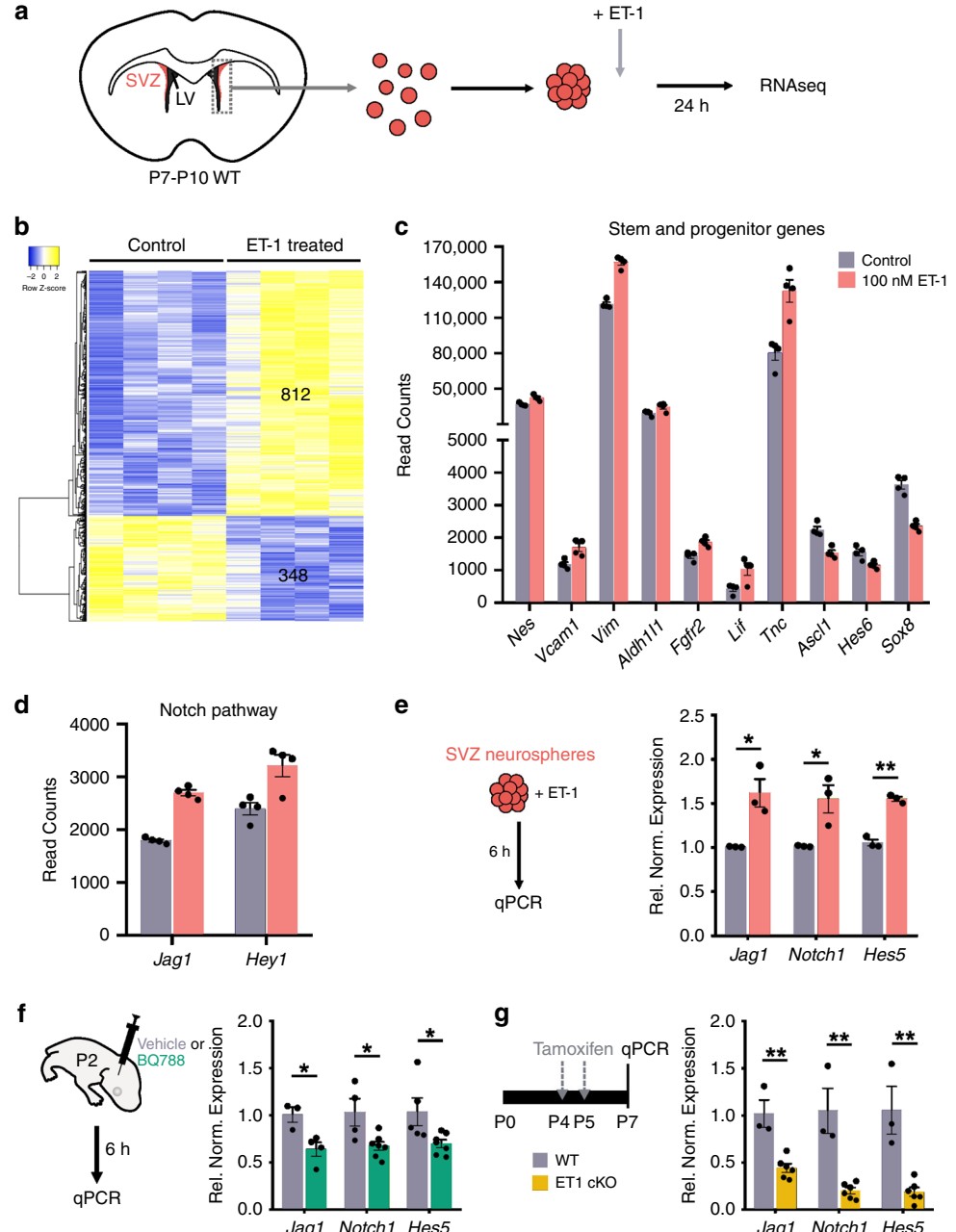

**Fig. 4 ET-1 regulates radial glial proliferation via Notch activation. a** Experimental procedure. **b** Heatmap depicting significant differentially expressed genes (DEGs) between control and ET-1-treated neurospheres. **c** Quantification of read counts of stem and progenitor genes that were significantly altered by ET-1 treatment (*n* = 4 independent batches). **d** Quantification of read counts of Notch pathway genes that were significantly altered by ET-1 treatment (*n* = 4 independent batches). **e** QPCR for Notch pathway components 6 h after ET-1 treatment (*n* = 3 independent batches). *p value = 0.036405; **p value = 0.001039 (Multiple *t*-tests with Holm–Sidak multiple comparisons correction). **f** QPCR for Notch pathway components in SVZ 6 h after stereotaxic injection of Ednrb antagonist BQ788 injection into LVs of P2 WT mice (*Jag1*: *n* = 3 saline and 4 BQ788 mice. *Notch1*: *n* = 4 saline and 7 BQ788 mice. *Hes5*: *n* = 5 saline and 7 BQ788 mice.). *p value = 0.048951 (Multiple *t*-tests with Holm–Sidak multiple comparisons correction). **g** QPCR for Notch pathway components in SVZ of WT and ET-1 cKO pups at P7 (*n* = 3 WT, 6 ET-1 cKO mice). **p value = 0.003901 (Multiple *t*-tests with Holm–Sidak multiple comparisons correction). Data are presented as mean values ± SEM. Source data are provided as a Source Data file.

that ET-1 activates different signaling pathways in these two progenitor populations and that loss of ET-1 signaling increases postnatal neurogenesis, thereby altering the cellular output of the SVZ and the ratio of glia to neurons. Finally, we demonstrate that ET-1 also regulates NSC and OPC proliferation in the adult mouse SVZ following focal demyelination of the SCWM, and that high levels of ET-1 correlate with an increased number of glia in

the SVZ of CARASAL patients. Together, our results identify ET-1 as a significant regulator of SVZ cell commitment and gliogenesis with therapeutic potential for promoting SVZ-mediated cellular repair after white matter injury or disease.

We found that ET-1 and its receptor Ednrb are both expressed by the majority of RGCs in the developing postnatal SVZ. While Ednrb expression by mouse and human RGCs has been described

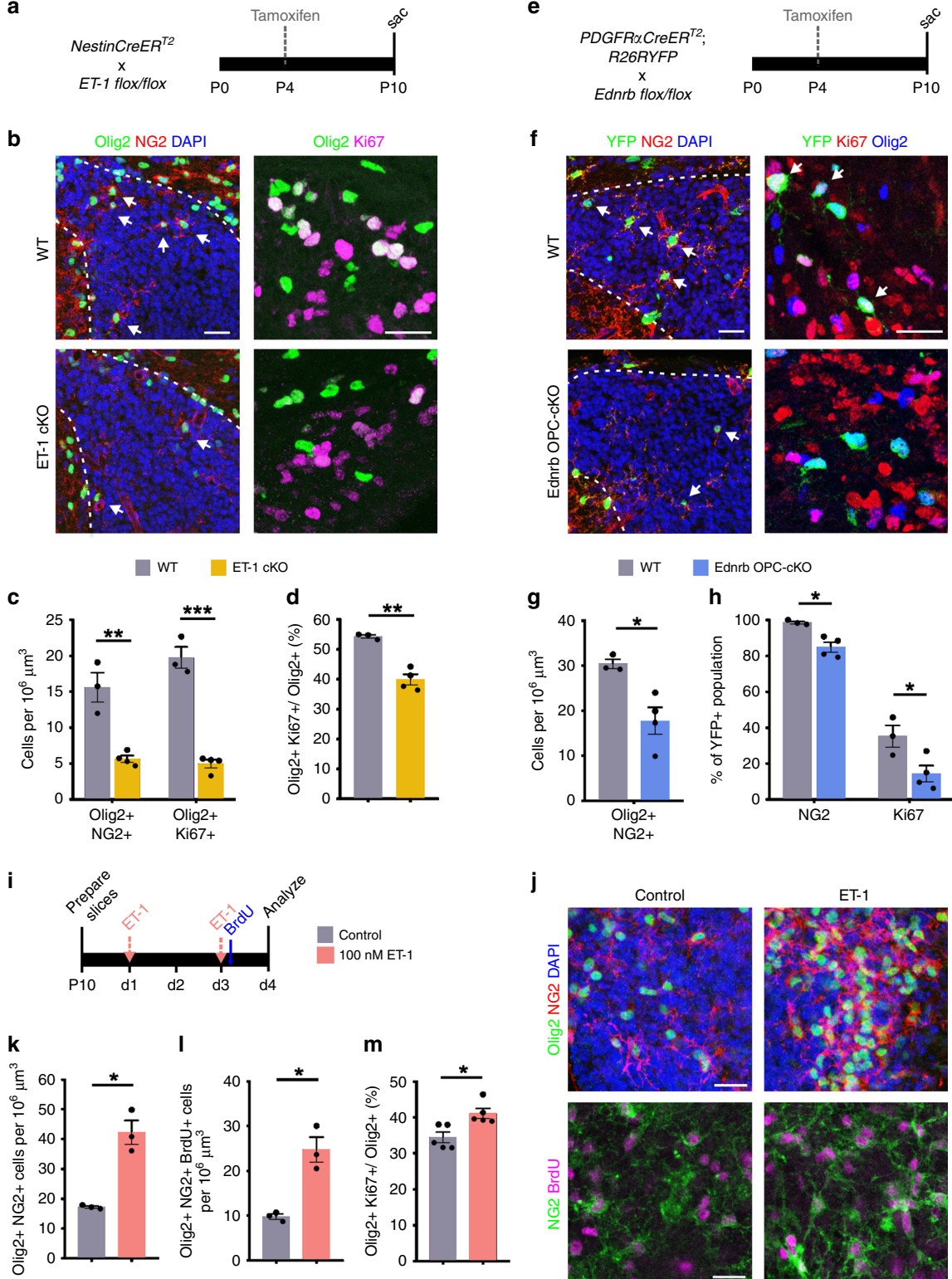

in several studies[35,36], its functional role within this cell type remains unknown. We found that ablation of either ET-1 or Ednrb in the early postnatal SVZ reduced the total number of RGCs at both short- (P10) and long-term (P28) timepoints. We demonstrated that this reduction was due to decreased RGC proliferation and increased differentiation of RGCs to NPCs—specifically the Sp8+ population. The *NestinCreER^{T2}* mouse strain largely induced recombination in the dorsal SVZ, where the majority of NPCs express Sp8. Sp8 is also expressed by most

migrating NPCs in the rostral migratory stream and remains expressed in the Calretinin+ and the nondopaminergic subpopulation of the GABAergic interneurons[37]. Whether ablation of ET-1 signaling in the ventral SVZ promotes increased neurogenesis of ventral NPCs, such as the Nkx2.1+ population, will be important to establish in future investigations. Notably, our neurosphere RNAseq screen revealed that ET-1 treatment induces significant upregulation of a stem cell gene network and downregulation of cell differentiation genes, including *Ascl1*.

**Fig. 5 ET-1 directly promotes OPC proliferation in the SVZ. a** Strategy for conditional and inducible ablation of ET-1 in the postnatal SVZ. **b** Coronal sections of WT and ET-1 cKO SVZ (outlined in white) at P10. **c** Quantification of the number of SVZ OPCs at P10 ($n = 3$ WT, 4 ET-1 cKO mice). **$p$ value = 0.002664; ***$p$ value = 0.000278 (Multiple $t$-tests with Holm–Sidak multiple comparisons correction). **d** Quantification of the percentage of proliferating SVZ OPCs at P10 ($n = 3$ WT, 4 ET-1 cKO mice). **$p$ value = 0.0026 (Welch's $t$-test). **e** Strategy for conditional and inducible ablation of Ednrb in OPCs. **f** Coronal sections of WT and Ednrb OPC-cKO SVZ at P10. **g** Quantification of total OPCs in the SVZ at P10 ($n = 3$ WT, 4 Ednrb OPC-cKO mice). *$p$ value = 0.0185 (Welch's $t$-test). **h** Quantification of the percentage of NG2+ and Ki67+ YFP+ OPCs in the SVZ at P10 ($n = 3$ WT, 4 Ednrb OPC-cKO mice). *$p$ value = 0.019675 (NG2); *$p$ value = 0.037353 (Ki67) (Multiple $t$-tests with Holm–Sidak multiple comparisons correction). **i** Organotypic slice assay. **j** Images of OPCs in the SVZ of control and ET-1-treated slices. **k** Quantification of total NG2+ OPCs in the SVZ following ET-1 treatment ($n = 3$ independent batches). *$p$ value = 0.0241 (Welch's $t$-test). **l** Quantification of total proliferating OPCs in the SVZ following ET-1 treatment ($n = 3$ independent batches). *$p$ value = 0.0289 (Welch's $t$-test). **m** Quantification of the percentage of proliferating OPCs in the SVZ following ET-1 treatment ($n = 5$ independent batches). *$p$ value = 0.0109 (Welch's $t$-test). All scale bars = 25 μm. Data are presented as mean values ± SEM. Source data are provided as a Source Data file.

Interestingly, our OPC RNAseq analysis identified significant downregulation of the immature neuronal maker *Dcx* following ET-1 treatment. These results suggest that ET-1 blocks a pan-neuronal fate in both RGCs and OPCs, shifting the SVZ towards gliogenesis. Therefore, the high expression of ET-1 and Ednrb within the early postnatal SVZ is likely a strong contributing factor in the dorsal wave of oligodendrogenesis that occurs during this window.

We previously found that ET-1 inhibits OL maturation and remyelination in the SCWM of the adult mouse brain following focal demyelination[15,16]. Our current study reveals that ET-1 also promotes OPC maintenance and proliferation in the SVZ, both during early postnatal development and following demyelination. Surprisingly, ET-1's effect on OPC proliferation appears to be restricted to the SVZ, as Ednrb-null OPCs did not display differences in proliferation in the CC at P14. While the mechanism underlying this region-specific effect of ET-1 signaling on OPC proliferation remains unknown, it may be due to regional differences between OPCs—as has been previously described for gray and WM OPCs[38]. Interestingly, single-cell sequencing of forebrain and spinal cord OPCs identified three distinct OPC populations, all of which express *Ednrb* at differing levels[39]. It is also possible that ET-1 signaling is modulated by other as yet unidentified pro-progenitor signaling molecules within the SVZ.

While ET-1 promotes proliferation of both RGCs and OPCs within the SVZ, we found that distinct signaling pathways are activated in each cell type. Using a neurosphere assay and two independent approaches to reduce ET-1 signaling in vivo, we first found that ET-1 induces upregulation of several Notch pathway components in RGCs, including *Jag1*, *Notch1*, and *Hes5*. This is in agreement with our previous finding that ET-1 signaling increases Jagged1 expression in reactive astrocytes following focal demyelination[15]. Notch signaling has been previously shown to promote and maintain RGC identity in the developing mammalian forebrain[24], as well as repress neuronal differentiation[25]. Therefore, downregulation of Notch signaling in the ET-1 and Ednrb cKO mice is likely responsible for both the decrease in RGCs and the increase in postnatal neurogenesis observed at P10 and P28. However, we did not detect any significant changes in Notch signaling components following ET-1 treatment of OPCs, indicating that ET-1 signaling does not directly activate the Notch pathway in OPCs. Our previous studies of WM demyelination found that Notch signaling was activated indirectly in OPCs via upregulation of Jagged1 in reactive astrocytes, with little direct activation of the Ednrb receptor on OPCs[15,16]. This is likely a result of the high number of reactive astrocytes present in SCWM lesions, thereby leading to increased lateral inhibition of OPC differentiation. In contrast, during development ET-1 signals largely via the Ednrb receptor directly on OPCs to regulate their proliferation and differentiation.

Injury to the CNS induces the transition of quiescent adult NSCs in the SVZ to actively dividing NSCs, generating new neuronal and glial progenitors that migrate to the damaged tissue[40]. Demyelination of the SCWM has been shown to stimulate SVZ gliogenesis, with newly generated OPCs migrating from the SVZ to lesions[30–32,41]. Interestingly, these SVZ-derived OPCs have been shown to produce thicker myelin sheaths than parenchymal OPCs[41]. Therefore, increasing SVZ oligodendrogenesis is a promising potential therapeutic avenue for demyelinating injury and disease. However, relatively little is known regarding the specific signals that regulate gliogenesis in the adult SVZ, as the majority of studies have focused on neurogenesis. We found that ET-1 is required for the increase in both NSC and OPC proliferation in the SVZ following focal demyelination of the SCWM. Importantly, we also found that high levels of ET-1 in the adult SVZ of CARASAL patients correlated with an increase in OPC number. This work has several important implications: First, that the functional role of ET-1 in promoting the proliferation and maintenance of glial progenitors is conserved in the human SVZ. Second, that ET-1 signaling could be potentially targeted to promote adult SVZ-mediated regeneration in neurodegenerative diseases, like Multiple Sclerosis, that result in extensive demyelination. This is in stark contrast to previous findings that blocking reactive astrocyte-derived ET-1 promotes repair after injury or disease[15,16]. Therefore, ET-1 levels will likely have to be carefully manipulated both spatially and temporally following disease or injury to result in optimal regenerative outcomes.

## Methods

**Mice.** *NestinCreER^{T2}* (Jackson ID #016261), floxed *Ednrb* (Jackson ID #011080), *PdgfraCreER^{T2}* (Jackson ID #018280), Rosa26YFP (Jackson ID #006148), *GFAP-CreER^{T2}* (Jackson ID #012849), and C57bl/6n (Jackson ID #000664) mice were purchased from the Jackson Laboratory. Floxed ET-1 mice were obtained from Dr. Ralph Shohet at the University of Hawaii as previously described[15]. All mouse colonies were maintained in the animal facility of Children's National Hospital, under standard conditions (65–75 °C, 40–60% humidity, 12 h light/dark cycles). All animal procedures were performed ethically, and according to the Institutional Animal Care and Use Committee of the Children's National Hospital (protocol #30578) and the Guide for the Care and Use of Laboratory Animals (National Institutes of Health).

**Tamoxifen and BrdU Injections.** Tamoxifen (Sigma) was dissolved in 100% ethanol and then diluted in sunflower oil (Sigma) to a final concentration of 10 mg/ml. For analysis of mutants at P10-11, one single tamoxifen injection (1 mg) was administered by IP at P4 (except for Fig. 4g). For analysis of mutants at P14 or older, 75 mg/kg tamoxifen was administered to nursing dams once per day for 3 consecutive days when pups were P3–P5. BrdU (Sigma, 10 mg/ml) was administered to pups by IP injection at P8 and P9 (10 μl per gram of pup) for analysis at P10.

**Tissue processing and Immunohistochemistry (IHC).** Mice were intracardially perfused and brains postfixed with 4% PFA. Tissue was cryoprotected with 30% sucrose in PBS. P10 brains were then embedded in OCT and frozen on dry ice. Sections (12 μm) were prepared on a cryostat, blocked in 10% donkey serum with 0.1% Triton X-100, incubated with primary antibodies diluted in blocking solution

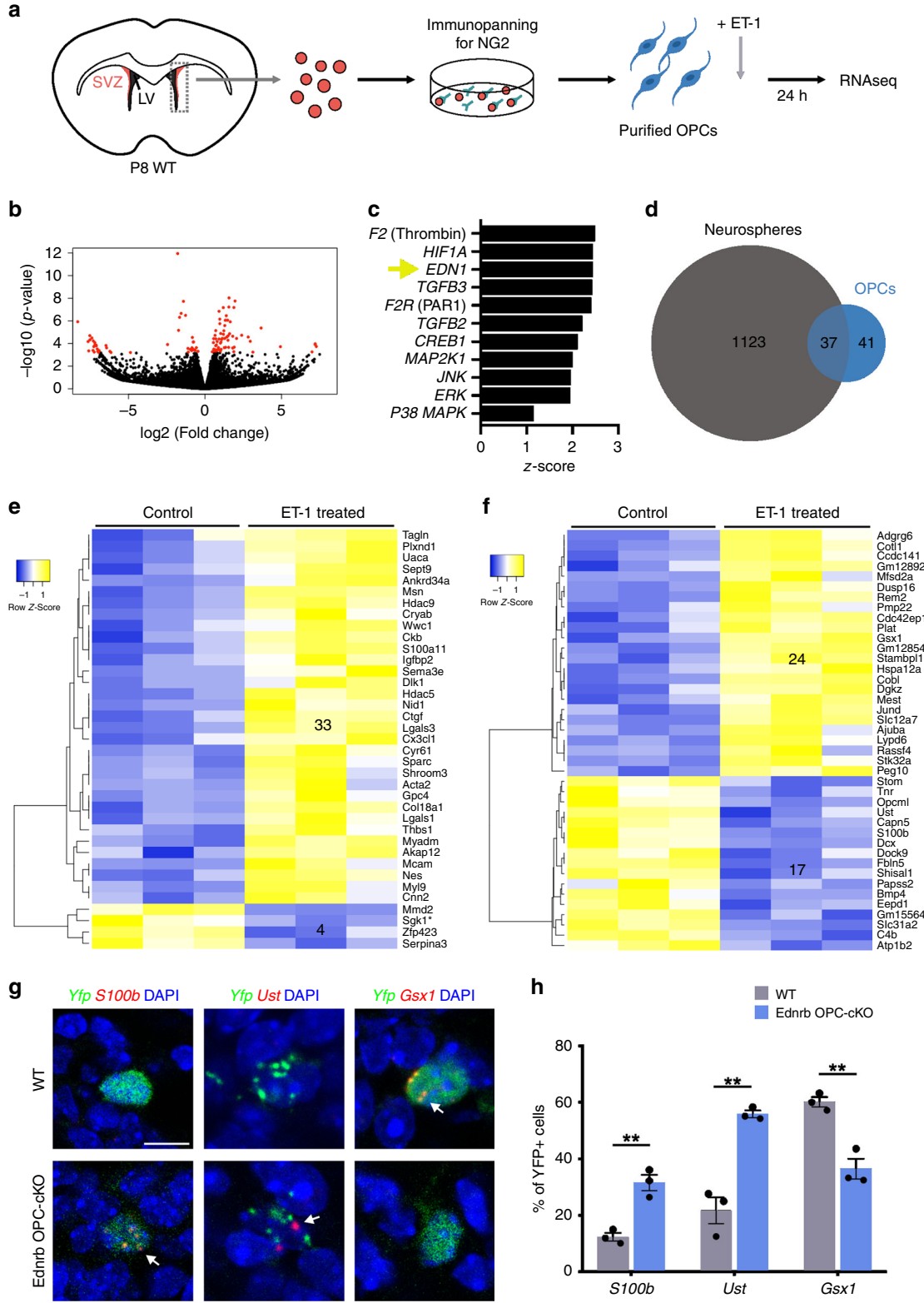

overnight at 4 °C, and incubated with appropriate fluorophore-conjugated secondary antibodies (Jackson) in blocking solutions at room temperature for 2 h. Brains older than P10 were sectioned using a freezing microtome with sections collected in PBS + 0.01% NaAz. Sections (40 μm) were blocked in 10% donkey serum with 0.1% Triton X-100, incubated with primary antibodies diluted in blocking solution for 1–2 days at 4 °C, and incubated with appropriate fluorophore-conjugated secondary antibodies at room temperature for 2 h. All sections were mounted with DAPI mounting media (Southern Biotech) and coverslipped.

**SVZ whole mount dissection and IHC**. SVZ whole mounts were dissected and stained as previously described[42]. The striatal wall of the LV was dissected from P11 pups in cold L-15 Leibovitz media, fixed with cold 4% PFA overnight, and washed with PBS with 0.1% Triton X-100 before staining. Whole mounts were blocked in 10% donkey serum with 2% Triton X-100 for 1 h at room temperature, and incubated with primary and secondary antibodies for 48 h at 4 °C. Whole mounts were then further dissected to isolate only the lateral wall, which was mounted with DAPI mounting media (Southern Biotech) and coverslipped.

**Fig. 6 Identification of downstream molecular targets of ET-1 signaling in OPCs. a** Experimental procedure. **b** Volcano plot displaying differentially expressed genes (DEGs) between control and ET-1-treated OPCs. The *y*-axis corresponds to the mean expression value of log10 (*p* value) and the *x*-axis displays the log2 fold change value. The red dots represent the transcripts that have a minimum log2 fold change of 0.05 and an adjusted *p* value of <0.1 (Wald test with Benjamini−Hochberg post hoc). **c** Predicted upstream regulators from IPA analysis of DEGs that are highlighted in red in **b**. Yellow arrow points to Edn1 as a predicted upstream regulator. **d** Venn diagram displaying the number of shared and distinct DEGs identified from the neurosphere and OPC RNAseq datasets. **e** Heatmap of the 37 DEGs shared by the neurosphere and OPC RNAseq datasets. The heatmap depicts the read counts from the OPC RNAseq data. Only 1 gene (Sgk1*) exhibited the opposite change in the neurosphere RNAseq dataset. **f** Heatmap of the 41 DEGs specific to the OPC RNAseq dataset. **g** Representative images of *Yfp*+ recombined OPCs in the dorsal SVZ of WT and Ednrb OPC-cKO mice at P10. Arrows point to the presence of *S100b, Ust,* or *Gsx1* mRNA puncta within the *Yfp*+ cells. Scale bar = 10 μm. **h** Quantification of the percentage of *Yfp*+ OPCs that contain *S100b, Ust,* or *Gsx1*+ mRNA puncta within the dorsal SVZ of WT and Ednrb cKO mice at P10. (*n* = 3 WT, 3 Ednrb OPC-cKO mice). **p** value = 0.007214 (*S100b* and *Gsx1*); **p** value = 0.006335 (*Ust*) (Multiple *t*-tests with Holm–Sidak multiple comparisons correction). Data are presented as mean values ± SEM. Source data are provided as a Source Data file.

**Antibodies**. The following primary antibodies were used for staining mouse tissue: rabbit anti-ET-1 (Abcam ab117757, 1:500), rabbit anti-Ednrb (Abcam ab117529, 1:500), rabbit anti-Ednra (Abcam ab117521, 1:500), rabbit anti-Olig2 (Millipore AB9610, 1:500), guinea pig anti-Olig2 (gift from Bennett Novitch), chicken anti-GFAP (Abcam ab4674 1:500), rabbit anti-S100β (Abcam ab868, 1:500), rabbit anti-BLBP (Millipore ABN14, 1:300), goat anti-Sox2 (Santa Cruz sc-17320, 1:100), guinea pig anti-Dcx (Millipore AB2253, 1:500), rabbit anti-NG2 (Millipore AB5320, 1:200), rat anti-BrdU (Abcam ab6326, 1:250), rabbit anti-Ki67 (Abcam ab16667, 1:200), rabbit anti-Ki67 (Vector labs VP-K451, 1:200), chicken anti-GFP (Aves labs GFP-1010, 1:1000), rabbit anti-Pax6 (Millipore AB2237, 1:500), rabbit anti-Gsx2 (Millipore ABN162, 1:300), goat anti-Sp8 (Santa Cruz sc-104661, 1:100), mouse anti-CC1/APC (Calbiochem OP80, 1:250), mouse anti-MBP (Covance SMI-99P, 1:500), rat anti-VCAM1 (BD Pharmingen 550547, 1:300), mouse anti-βcatenin (BD Biosciences 610154, 1:500), rabbit anti-cleaved Caspase3 (Cell Signaling 9664S, 1:200), mouse anti-NeuN (Millipore MAB377, 1:500), mouse anti-Mash1 (BD Pharmingen 556604, 1:200), chicken anti-Nestin (Aves labs NES, 1:500), rat anti-CD31 (BD Pharmingen 550274, 1:50), goat anti-Sox9 (R&D Systems AF3075, 1:500), rat anti-PDGFRa (BD Pharmingen 558774, 1:250), and rabbit anti-Aldh1L1 (Abcam ab87117, 1:250). The species-appropriate Alexa488, Alexa594, and Alexa647 secondary antibodies (Jackson Immunoresearch, 1:500) were used.

**Quantitative PCR**. Total RNA was isolated from microdissected brain tissue using the RNeasy lipid tissue mini kit (Qiagen). Synthesis of cDNA was carried out using the iScript Reverse Transcription Supermix for RT-qPCR (Biorad). qPCR was performed on a CFX96 real-time system (Biorad) in a 20 µl reaction mixture using SsoAdvanced Universal SYBR Green PCR master mix (Biorad). Cycle parameters were 10 s at 95 °C and 30 s at 60 °C. Data were normalized to GAPDH. Primer sequences are listed in Supplementary Tables 1 and 2.

**In situ hybridization (ISH)**. ISH was performed on C57bl/6n brain sections (12 µm) as previously described[43]. DIG-labeled probes were generated by PCR using cDNA generated from P14 C57bl/6n whole-brain RNA. Sections were postfixed with 4% PFA for 10 min, followed by three, 5-min washes in PBS, and a 10 min incubation in acetylation mix (0.1% triethanolamine and 0.02% acetic anhydride in water). After three more PBS washes, sections were incubated in warm hybridization buffer (50% formamide, 5× SSC, 5× Denhardts, 250 µg/ml yeast RNA, 500 µg/ml salmon sperm in water) for several hours at 68 °C, followed by an overnight incubation with 500 ng/ml probe in hybridization buffer. Next, sections had sequential hour-long incubations (68 °C) in formamide solution 1 (%50 formamide, 5× SSC, 1% SDS in water) and formamide solution 2 (%50 formamide, 2× SSC, 1% Tween20 in water). Following two, 10-min washes with 1% Tween20 in 1× Tris-buffered saline (TBST), sections were blocked with 10% heat-inactivated horse serum in TBST for 1 h at room temperature. Sections were then incubated with anti-DIG antibody (1:2000) for 1.5 h at room temperature, before development with 0.02% NBT-BCIP (Roche) in alkaline phosphatase buffer.

**Neurosphere assay**. The SVZ was microdissected from P7-P10 C57bl/6n mouse pups in Leibovitz's L-15 medium on ice. SVZ tissue was enzymatically dissociated with Papain/DNase (Worthington) and single cells were cultured in serum-free medium at 10,000 cells/ml with 10 ng/ml FGF2 (Peprotech) and 10 ng/ml EGF (Invitrogen) in ultralow attachment six-well plates (Costar) for 7 days. Serum-free medium consisted of DMEM/F12, N-2, B-27, and Pen/Strep (all Invitrogen). Neurospheres were then used for one of several experiments: (1) For RNAseq: 100 nM ET-1 (Tocris) was added to the cells on day 4 and neurospheres were collected either 6 h (for qPCR) or 24 h later for RNA isolation. Total RNA was isolated from control and ET-1-treated neurospheres using the RNAeasy Plus Mini Kit (Qiagen) and stored at −80 °C. Four independent batches of neurospheres were generated and treated with ET-1 to represent our biological replicates for RNAseq. (2) For IHC analysis: 100 nM ET-1 was added to the cells on day 0, 3, and 5. On day 6, neurospheres were imaged, fixed in 4% PFA for 30 min on ice, cryoprotected

with 30% sucrose in PBS, and embedded in OCT. Twelve microns cryosections were then stained using standard IHC techniques described above. (3) For the BrdU assay: 100 nM ET-1 was added to the cells on day 0, 3, and 5. After 20 min of ET-1 treatment on day 5, BrdU (10 µM) was added to the cells. Control and ET-1-treated neurospheres were collected 4 h later for IHC analysis.

**Organotypic slice culture assay**. Organotypic cultures were prepared as previously described[44]. Brains from C57bl/6 mice (P9-P10) were dissected and glued (Super Glue Loctite) onto the chuck of an ice cooled vibratome Leica VT1200S. Two hundred and seventy-micron-thick coronal sections were cut and collected in chilled neurobasal medium enriched with N-2 supplement and 10 µg/ml gentamicin (all reagents Gibco/ThermoFisher). The coronal slices were cultured on Millicell Cell Culture Inserts (Millipore, pore size 0.4 µm, diameter 30 mm) in media consisting of 50% MEM/HEPES, 25% HBSS, 25% horse serum, 2 mM NaHCO₃, 6.5 mg/ml glucose, 2 mM glutamine (all reagents Gibco/ThermoFisher), and 100 µg/ml primocin (InvivoGen). Organotypic slices were cultured for 96 h. The media were replaced after 1 and 3 days in vitro, during which 100 nM ET-1 (Tocris) was added to randomly selected groups of slices. BrdU (10 µM) was added to all slices on day 3 of culture for ~16 h incubation prior to fixation. Organotypic slices were fixed for 3 h at 4 °C in 4% PFA and then stored at 4 °C in PBS with 0.01% NaAz until use. Organotypic brain slices were blocked for 4 h in PBS with 0.1% Triton X-100 (PBST) and 10% donkey serum. The slices were then incubated with primary antibodies in PBST with 1% normal donkey serum 2 days overnight at 4 °C with orbital shaking. Slices were then washed in PBST four times for 30 min and incubated with species specific fluorescently labeled secondary antibodies overnight at 4 °C. The day after, slices were washed with PBST three times for 30 min, incubated with DAPI for 30 min (Sigma-Aldrich), and were coverslipped with Fluoromont-G mounting media (Southern Biotech). For BrdU staining: after fixation, slices were incubated in 2N HCl for 30 min followed by washing in borate buffer (pH 8.5) for 30 min.

**In vivo peptide injections**. P2 C57bl/6n mouse pups were anesthetized on ice. Two microlitres of vehicle or BQ788 (Tocris, 300 pmol) were injected into each LV using a Hamilton Syringe. BQ788 is a selective antagonist of the Ednrb receptor[45]. The LVs were targeted by injecting at 2/5 of the distance from the lambda suture to each eye, at a depth of ~3 mm. CellTracker Green CMFDA (Molecular Probes) was used to label injection site to confirm correct targeting. Following injections, mouse pups recovered on a warming pad until body temperature, skin color, and movement returned to normal. Pups were then returned to their biological mother. For qPCR: pups were sacrificed 6 h after injections via cervical decapitation and the SVZ was microdissected for total RNA isolation. For IHC analysis: pups were sacrificed 48 h after injections via cervical decapitation and brains were drop-fixed in 4% PFA overnight at 4 °C. No toxicity or side effects of BQ788 injection were observed.

**OPC immunopanning**. OPCs were purified via immunopanning and cultured in vitro as previously described[46]. The SVZ was microdissected from P7-P8 C57bl/6n mouse pups in Leibovitz's L-15 medium on ice. SVZ tissue was enzymatically dissociated with Papain/DNase (Worthington) into single cells and OPCs were positively selected via incubation on a NG2 antibody-coated petri dish for 45–60 min. Nonbound cells were washed off the dish and bound-OPCs were removed via trypsinization. OPCs were plated on PDL-coated six-well plates in DMEM-SATO base growth medium with B-27 (Invitrogen), forskolin (4.2 µg/ml), CNTF (10 ng/ml), PDGF (10 ng/ml), and NT-3 (1 ng/ml). Media were changed every 2 days. The purity of the cultures was assessed by staining for Olig2 and PDGFRα. Overall, 100 nM ET-1 was added to half of the cells after 3–4 days in vitro and cells were harvested 24 h later. Total RNA was isolated from control and ET-1-treated OPCs using the RNAeasy Plus Mini Kit (Qiagen) and stored at −80 °C. Three independent batches of OPCs were generated and treated with ET-1 to represent our biological replicates.

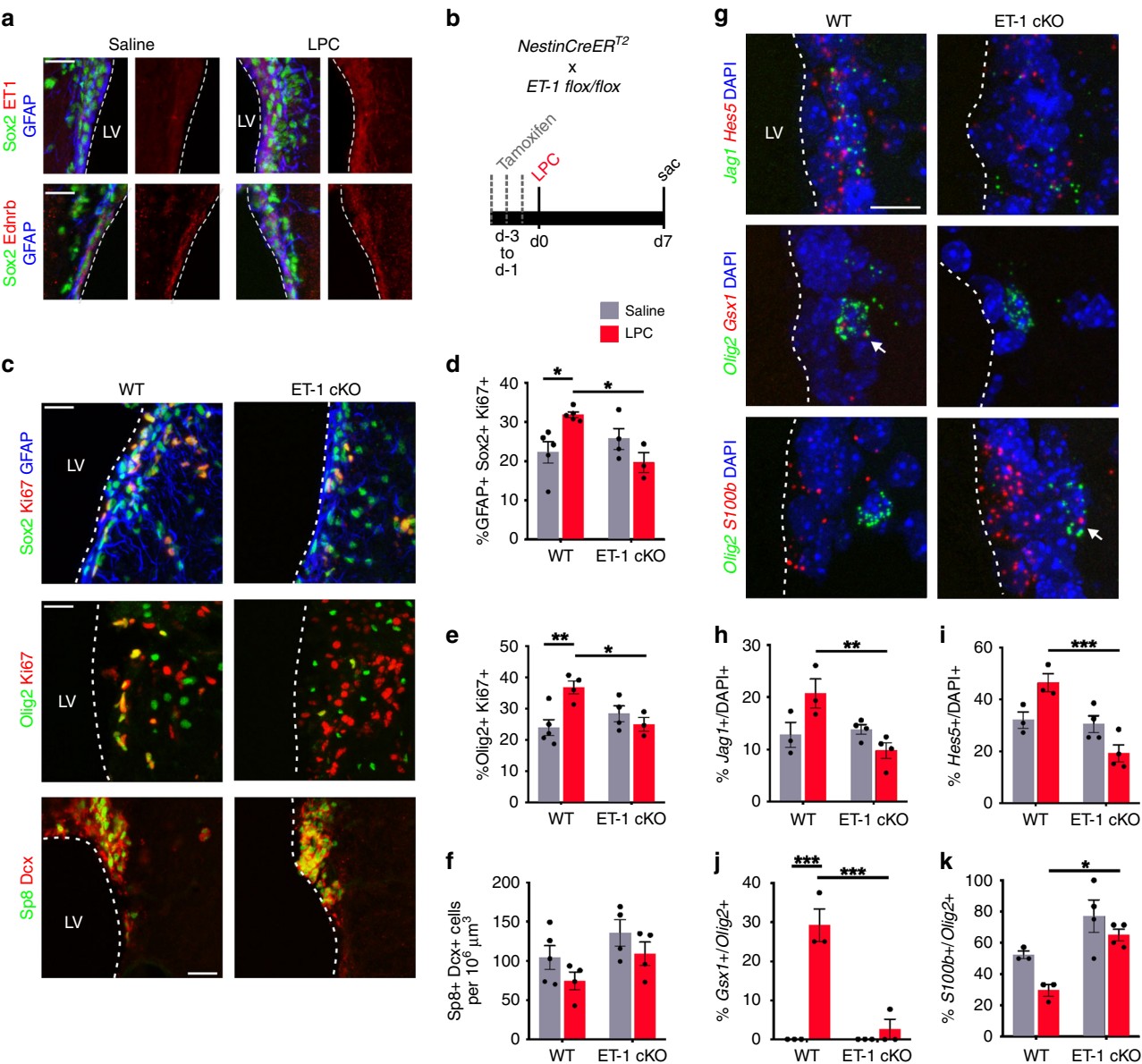

**Fig. 7 ET-1 signaling is reactivated in the adult mouse SVZ following demyelination. a** Expression of ET-1 and Ednrb in the adult SVZ of WT mice 7 days following injection of saline or lysolecithin (LPC) into the SCWM. Images are representative of three WT mice. Scale bar = 25 μm. **b** Experimental strategy. **c** Coronal sections of adult WT and ET-1 cKO SVZ 7 days following LPC injection. Scale bar = 25 μm. **d** Quantification of the percentage of proliferating NSCs in the SVZ (Saline: $n = 5$ WT, 5 ET-1 cKO mice. LPC: $n = 5$ WT, 3 ET-1 cKO mice). *$p$ value = 0.0284 (WT saline versus WT LPC); *$p$ value = 0.0156 (WT LPC versus ET-1 cKO LPC) (two-way ANOVA with Tukey's multiple comparisons test). **e** Quantification of the percentage of proliferating OPCs in the SVZ (Saline: $n = 5$ WT, 4 ET-1 cKO mice. LPC: $n = 4$ WT, 3 ET-1 cKO mice). **$p$ value = 0.0097; *$p$ value = 0.0364 (two-way ANOVA with Tukey's multiple comparisons test). **f** Quantification of Sp8+ Dcx+ NPCs in the SVZ (Saline: $n = 5$ WT, 4 ET-1 cKO mice. LPC: $n = 4$ WT, 4 ET-1 cKO mice). **g** RNAscope labeling of coronal sections of adult WT and ET-1 cKO SVZ 7 days following LPC injection. Arrows point to red puncta within *Olig2*+ cells. Scale bar = 10 μm. **h** Quantification of the percentage of *Jag1*+ cells in the dorsal SVZ ($n = 3$ WT, 4 ET-1 cKO mice). **$p$ value = 0.0086 (two-way ANOVA with Tukey's multiple comparisons test). **i** Quantification of the percentage of *Hes5*+ cells in the dorsal SVZ ($n = 3$ WT, 4 ET-1 cKO mice). ***$p$ value = 0.0009 (two-way ANOVA with Tukey's multiple comparisons test). **j** Quantification of the percentage of *Gsx1*+ OPCs in the dorsal SVZ ($n = 3$ mice both groups). ***$p$ value = 0.0001 (WT saline versus WT LPC); ***$p$ value = 0.0003 (WT LPC versus ET-1 cKO LPC) (two-way ANOVA with Tukey's multiple comparisons test). **k** Quantification of the percentage of *S100b*+ OPCs in the dorsal SVZ ($n = 3$ WT, 4 ET-1 cKO mice). *$p$ value = 0.0187 (two-way ANOVA with Tukey's multiple comparisons test). LV lateral ventricle. Data are presented as mean values ± SEM. Source data are provided as a Source Data file.

**RNAseq library preparation and sequencing**. The RNA samples were sent to the Penn State Hershey Genome Sciences and Bioinformatics Facility at Penn State College of Medicine. The RNAseq libraries were prepared using the NEXTflex Rapid Directional RNA-Seq Kit. The libraries were indexed (Unique Dual Index Barcodes) and pooled together for sequencing on an Illumina NovaSeq 6000 using paired-end reads of 50 bp. The raw sequences were de-multiplexed based on the index sequence and trimmed.

**RNAseq analysis**. The quality of the raw fastq reads from the sequencer was evaluated using FastQC. HISAT2 (version 2.1.0)[47] was used to map the reads to the reference mouse genome (GRCm38/mm10). The output SAM files were sorted by coordinate and converted to BAM files using SAMtools (version 1.3)[48]. Duplication reads were marked using Picard (version 2.17.6) MarkDuplicates. The mapped reads were counted along with the multimapping and multioverlapping reads using featureCounts (subread version 1.6.2)[49] with a reference genomic feature file (Gene

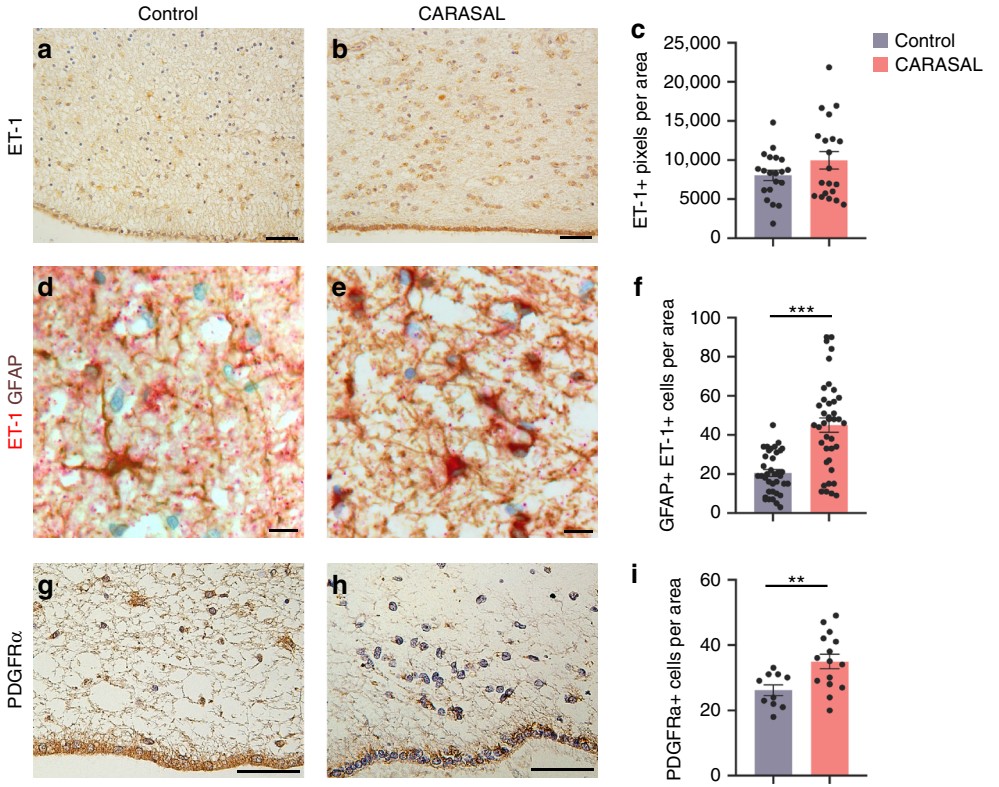

**Fig. 8 ET-1 protein is increased in the SVZ of adult CARASAL patients. a, b** ET-1 staining in the adult human SVZ of CARASAL patients and age-matched controls. Scale bars = 50 μm. **c** Quantification of the number of ET-1+ pixels in the SVZ ($n = 20$ images of the SVZ of two control and two CARASAL patients). **d, e** ET-1+ GFAP+ astrocytes in the SVZ of CARASAL patients, compared with controls. Scale bars = 25 μm. **f** Quantification of the number of GFAP+ ET-1+ cells per area in the SVZ ($n = 37$ images of the SVZ of two control patients, 39 images of the SVZ of two CARASAL patients). ***$p$ value < 0.0001 (Welch's $t$-test). **g, h** PDGFRα staining in the adult human SVZ of CARASAL patients and age-matched controls. Scale bars = 50 μm. **i** Quantification of the number of PDGFRα+ cells per area in the SVZ ($n = 10$ images of the SVZ of two control patients, 15 images of the SVZ of two CARASAL patients). **$p$ value = 0.0044 (Welch's $t$-test). Data are presented as mean values ± SEM. Source data are provided as a Source Data file.

transfer format) (mm38). Differential gene expression analysis was performed using default parameters with Deseq2 (version 1.22.2)[50]. For the neurosphere RNAseq dataset, the threshold for defining significant DEGs was an adjusted $p$ value < 0.05. For the OPC RNAseq dataset, the threshold for defining significant DEGs was an adjusted $p$ value < 0.1. The volcano plot was generated in R and heatmaps were generated using Heatmapper (http://www.heatmapper.ca/). Selected genes were then validated via qPCR or RNAscope.

**Focal demyelination of the adult mouse SCWM**. Seventy-five micrograms per kilogram tamoxifen were administered to adult (8–12 weeks old) mice daily for 3 consecutive days. The following day, mice were deeply anesthetized using 100 mg/kg ketamine and 10 mg/kg xylazine. Lysolecithin (1% LPC, 2 μl, EMD Chemicals) was injected unilaterally into the SCWM of mice using a Hamilton syringe. On the contralateral side, 2 μl 0.9% NaCl was injected for control purposes. Injections were made using a stereotaxic apparatus at the following coordinates: 1.0 mm anterior to bregma, 1.5–2 mm lateral, and 3.0 mm deep. Mice recovered for 7 days post injection and were subsequently perfused for IHC or RNAscope analysis.

**RNAscope *in situ* hybridization**. Tissue was collected and processed using standard procedures described above. Twelve microns cryosections were collected and stored at −80 °C prior to use. The RNAscope Multiplex Fluorescent Reagent Kit V2 (323100) by Advanced Cell Diagnostics was used to perform fluorescent labeling of different RNA molecules according to the kit manual, with the following changes: $H_2O_2$ pretreatment for 5 min at room temperature, followed by washes in distilled water and 100% ethanol for 3 min. No target antigen retrieval step was performed. Slides were then dried at room temperature. RNAscope probes used were: *Yfp* (312131), *Gsx1* (571501), *S100b* (431731), *Ust* (555891), *Jag1* (412831), *Hes5* (400991), and *Olig2* (447091). For the analysis, the dorsal SVZ was imaged using a Leica SP8 confocal microscope. Cells with at least two fluorescent puncta were counted as positive for that probe.

**Image acquisition and analysis**. Mouse brain sections, SVZ whole mounts, and organotypic slices were imaged with a Zeiss LSM 510 or a Leica SP5 confocal laser-scanning microscope. SVZ whole mount fields were random fields selected from

anterior–dorsal areas. All images were processed and quantified using Fiji ImageJ and CorelDraw software.

**Human tissue processing and analysis**. Paraffin-embedded (5 μm) sections from CARASAL and control patients were deparaffinized in xylene, rehydrated in descending grades of alcohol and washed in PBS. Endogenous peroxidase activity was blocked with 0.3% $H_2O_2$ in PBS for 30 min. For antigen retrieval, slides were heated in a microwave oven for 10 min in 0.01 M citrate buffer (pH 6.0). After antigen retrieval, sections were allowed to cool to room temperature and rinsed in PBS. Sections were incubated with a blocking solution of 5% normal goat serum in PBS for 1 h. Sections were then incubated for 2 h with anti-GFAP antibody (Z0334, Dako, 1:1000) or anti-PDGFRα antibody (sc-12911, Santa Cruz). After washing, sections were incubated with Envision mouse/rabbit HRP (Dako) for 1 h. Sections incubated with HRP were visualized with DAB (Dako). For double IHC, sections were then incubated with anti-ET-1 antibody (H54085m, Meridan, 1:100) overnight. After washing, sections were incubated with an alkaline phosphatase-labeled secondary antibody for 1 h, washed in Tris-buffered saline, and stained using Liquid Permanent Red (Dako). Sections were counter-stained with haematoxylin and mounted with Aquatex (Millipore). All incubation steps were performed at room temperature.

Images were taken with a Leica DC500, and the number of positive ET-1 pixels was determined using the color deconvolution macro in ImageJ. For the number of positive PDGFRα positive cells per area, the cell counter tool of ImageJ was used. The total number of GFAP+ ET-1+ cells were counted in at least ten standardized fields using a ×10 objections lens by two observers.

**Statistics**. Specific numbers of animals or cell culture preparations are denoted in each figure legend. Data were compiled and organized using Microsoft Excel. All graphs and statistical tests were generated using Graphpad Prism 7.0 software. All data are presented as averages ± SEM. The statistical test used for each graph is stated in the figure legends. All statistical tests were two-sided and a two-tailed type 1 error ($p$ value < 0.05) was used to determine statistical significance. The degree of statistical significance was denoted using asterisks (*$p$ < 0.05; **$p$ < 0.01; ***$p$ < 0.001). Exact $p$ values are provided in the figure legends.

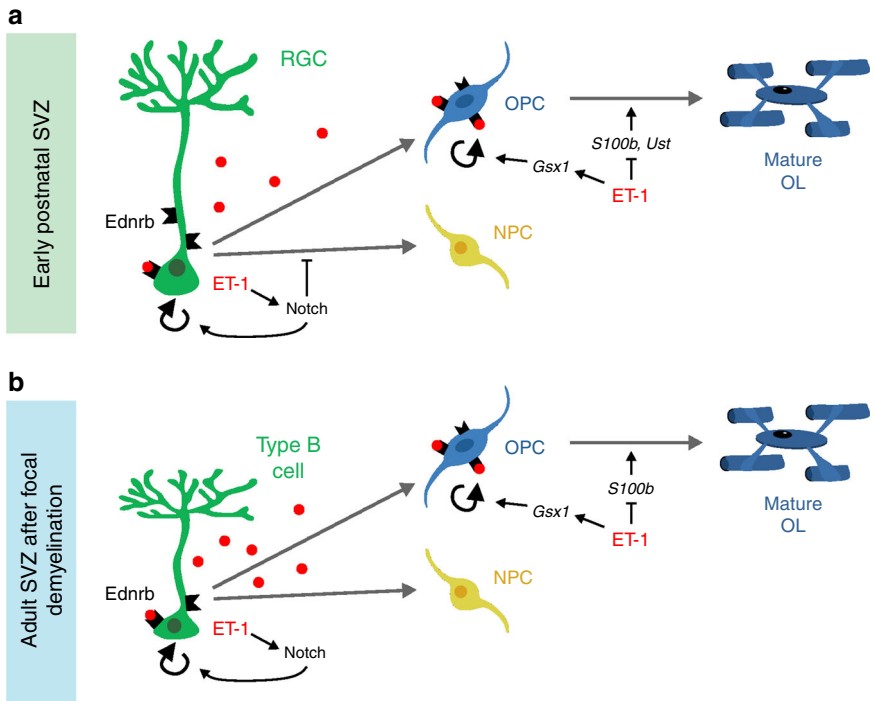

**Fig. 9 Model of ET-1 signaling in the postnatal SVZ. a** In the developing postnatal SVZ, RGCs secrete ET-1, which binds to Ednrb receptors on RGCs in an autocrine manner to activate Notch signaling. This leads to upregulation of stem cell genes and downregulation of proneural factors, resulting in RGC maintenance and proliferation. ET-1 also binds to Ednrb receptors on OPCs within the SVZ to upregulate the transcription factor *Gsx1* and downregulate OL maturation factors *S100b* and *Ust*, thereby promoting OPC proliferation and blocking maturation. **b** Following demyelination of the SCWM, Type B cells in the adult mouse SVZ upregulate ET-1. This induces upregulation of Notch signaling components *Jag1* and *Hes5* in Type B cells, leading to increased proliferation. ET-1 also induces upregulation of *Gsx1* and downregulation of *S100b* in SVZ OPCs, increasing their proliferation as well.

**Reporting summary**. Further information on experimental design is available in the Nature Research Reporting Summary linked to this paper.

## Data availability

All RNA-sequencing data have been deposited in the NCBI Sequence Read Archive (accession code: PRJNA607509). The authors declare that all data supporting the findings of this study are available within the paper and its Supplementary Information files. Source data for Figs. 1–8 and Supplementary Figs. 1–10 are provided as a Source Data file.

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

## Acknowledgements

We are grateful to members of the Gallo laboratory and Center for Neuroscience Research at Children's National Hospital for suggestions and comments, especially Drs Tom Forbes and Evan Goldstein. We thank Dr Bennett Novitch for providing the guinea pig anti-Olig2 antibody. This work was supported by Award numbers R01NS090383 from NINDS (V.G.), U54HD090257 from National Institute of Child Health and Human Development (NICHD) (District of Columbia Intellectual and Developmental Disabilities Research Center (DC-IDDRC)) (V.G.), and 5F32NS098647 from NINDS (K.L.A.). Microscopic analysis was carried out at the Children's National Research Institute (CRI) Cell and Tissue Microscopy Core, which is supported by DC-IDDRC grant U54HD090257 (NICHD). We also acknowledge the support of the CRI Bioinformatics Unit, a partnership between the CRI, the Center for Genetic Medicine Research, the Clinical Translational Science Institute at Children's National (CTSI-CN), and the DC-IDDRC. The CTSI-CN is supported through the National Institutes of Health (NIH) Clinical and Translational Science Award (CTSA) program, grant UL1TR001876 and KL2TR001877. The CTSA program is led by the NIH's National Center for Advancing Translational Sciences. The DC-IDDRC is supported through the NIH DC-IDDRC Award (DC-IDDRC) program, grant (1U54HD090257). The DC-IDDRC program is led by NIH, Eunice Kennedy Shriver NICHD. This content is solely the responsibility of the authors and does not necessarily represent the official views of the NIH.

## Author contributions

K.L.A. designed and conducted experiments and performed data analysis. G.R. performed and analyzed the organotypic brain slice cultures. P.B. analyzed the RNAseq data. Both M.B's stained and analyzed the human SVZ tissue samples. V.G. designed experiments and supervised the project. K.L.A. and V.G. wrote the manuscript.

## Competing interests

The authors declare no competing interests.
