## [Peer Review File · Nature Communications]

Reviewers' comments:

Reviewer #1 (Remarks to the Author):

This study by Adams et al. focuses on the impact of endothelin-1 on progenitor cells in the postnatal subventricular zone (SVZ). The authors conclude that endothelin-1 (ET-1) promotes radial glial cell maintenance and proliferation. Reduced ET-1 signaling increases neurogenesis and reduces oligodendrocyte progenitor cell (OPC) production and proliferation. They show that ET-1 is required for increased neural stem cell and OPC proliferation in the adult mouse SVZ after demyelination. To assess whether these data relate to human progenitor cell behavior, they studied Cathepsin A-related arteriopathy with strokes and leukoencephalopathy (CARASAL) patients. In at least one variant of this disorder, cathepsin protease activity is reduced, resulting in increased ET-1 levels. In those samples, there were increased SVZ OPCs, indicating that ET-1 also impacts SVZ progenitors in humans.

This laboratory has been investigating the impact of ET-1 on a number of cells following injury and in developing brain. This focus on the SVZ is a new and possibly quite important aspect of ET-1 function in the developing brain and after injury. As they note, few studies focus on the role of ET-1 in early development and SVZ specification. The data demonstrating signaling via ET-1 and its receptor at the SVZ are strong. The figures generally present strong and convincing data. The authors provide a model for endothelin signaling that impacts neuronal and oligodendrocyte specification during development and after injury.

While these are clear experiments, some issues should be addressed. The most crucial is the significance of this work, relative to their earlier work on the impact of ET-1 signaling in oligodendrocytes. In general, the data here are consistent with those from their earlier work. It is encouraging that they see comparable outcomes from loss or overexpression of the ET-1 signaling system, but it is not unexpected. The unique and most important aspect of this work is the focus on SVZ cells and outcomes with respect to neuronal vs oligodendrocyte lineage commitment. With respect to the latter part of the results, the overall conclusions are comparable to their earlier work. They note that their earlier work showed that ET-1 inhibits oligodendrocyte maturation and remyelination and suggest that contrasts with the current work showing that loss of ET-1 reduced OPC numbers and proliferation. This is not necessarily a contrasting observation, since ET-1 driving OPC proliferation would reduce their maturation and remyelination. If they directly compare OPC proliferation near the SVZ to that in medial or lateral corpus callosum in an injury state, does ET-1 impact OPCs differently? Seems unlikely, but is so, that would be important to demonstrate. They state that they did not see an impact of ET-1 on OPC proliferation in their earlier work, but as they note some of those data were in vitro studies and certainly in the Hammond et al Neuron, 2013 paper, there is a non-significant increase in OPCs after lysolecithin. As above, the unique aspect of this work is the impact of ET-1 on SVZ cell commitment, much more than the impact on OPC/oligodendrocyte behavior and emphasizing that more would enhance this paper. The other important element of this work is the demonstration that this response also occurs in human tissue. They have shown this in earlier studies in tissue from multiple sclerosis patients but highlighting and expanding both these elements would focus this paper to the concepts that make this study novel.

Specific comments:

The loss of ET-1 or Ednrb from SVZ radial glial cells reduces their proliferation and shifts their commitment to the neuronal lineage without impacting the number of Olig2+ or S100B+ cells. It would be helpful to provide discussion of how the reduced number of radial glial cells and the increased numbers of neurons occur without any impact on oligodendrocyte numbers. Additionally, do the SVZ radial glial cells continue to have reduced ET-1 or Ednrb by the time there are excess NeuN+ neurons in the olfactory bulb or do radial glial cells that escape deletion outcompete the poorly proliferating ET-1- or Ednrb-null cells. Do the excess neurons in the olfactory bulb

differentiate to functioning neurons or do they die in the bulb?

The authors demonstrate that following lysolecithin-induced demyelination in adults, there is normally increased proliferation of neural stem cells at the SVZ and an increase in OPCs. This has been seen in other contexts. One question is whether this is a shift from the normal production of neuronal precursors that move up the rostral migratory stream or rather an additional element of differentiation to the oligodendrocyte lineage from the increased number of SVZ precursors? Other studies show that after TBI, there are increased neuroblasts migrating through the rostral migratory stream, whereas MPTP reduces the number (Dixon et al., 2016; He and Nakayama, 2015). When damage to the olfactory bulb occurs, is ET-1 involved in the commitment to neuroblasts and their proliferation, or is this exclusively an impact of ET-1 on differentiation of SVZ progenitors to the oligodendrocyte lineage?

With respect specifically to the figures:

In Figure 1, it is very hard to distinguish individual cells stained for GFAP, S100b, BLBP, ET-1 or Ednrb. Thus, it is hard to assess how the numbers of percent of specific cell populations would be generated. Additionally, it is unclear what everything is compared to in A-C? B and C show changes over development, but as in A, the identification of what 1 represents should be defined in the figure legend.

Reviewer #2 (Remarks to the Author):

In this manuscript, the authors revealed the ET-1 signaling function in the early postnatal SVZ. Using cKOs of ET-1 and one of its receptor Ednrb, they showed that ET-1 promotes RGC proliferation via Notch signaling in an autocrine manner and OPC proliferation in a paracrine manner in normal development. They also showed ET-1 signaling is required for NSC and OPC proliferation in the adult SVZ after demyelination and suggested that such function is conserved in the human SVZ by examining the brain of the CARASAL patients.

This manuscript provides solid evidence regarding ET-1 functions in the postnatal SVZ. Their finding that ET-1 signaling promotes OPC proliferation in the SVZ is novel, and of note, it is interesting that loss of ET-1 signaling increases postnatal neurogenesis; however, the reviewer also thinks the overall findings of this manuscript are consistent with, and similar to the previous reports regarding the function of ET-1 signaling pathway in the pathological condition (Gadea et al., 2009, Hammond et al., 2014, 2015). It has been known that the RGCs and reactive astrocytes share similar characters; the RNAseq analysis in this manuscript regarding ET-1 signaling in OPCs further supports the similarity to the reactive gliosis. In this manuscript, the authors show that ET-1 signaling is conserved in the human SVZ in the pathological condition (CARASAL patients), but the involvement of ET-1 signaling in CARASAL has already reported in WM astrocytes (Bugiani et al., 2016), and the authors extended the examination to the SVZ glial progenitors. The authors discuss the dissimilarity between physiological SVZ and the pathological condition in the point of ET-1 function on OPC proliferation; however, the reviewer still wonders whether this manuscript would provide enough high impact on the research field.

Other points:

Figure 2J and 3C, H: Does the ET-1 cKO at the early postnatal ages also have the long term effect on the maintenance of the adult NSCs at P28, the fate of the YFP cells at P10, and neurogenesis at P28? These experiments would clarify whether the above effects are in an autocrine manner.

Figure 7D, E:

The authors should estimate ET-1 function in the adult SVZ under physiological condition by comparison of WT and ET-1cKO in a saline experiment.

Page3 line19: The explanation of abbreviation 'OL' is lacked.

Page5 lines7 and Fig2E:

No significant difference does not 'indicate' that ependymal cells were not affected. The statistically correct interpretation should be used to describe the results.

Do the VCAM1(-) cells in Fig2 B represent the ependymal cells? If so, their cell density may also provide information on this point.

In the bar plots throughout the manuscript, please overlay the dot plots because the sample size is small. Do the error bars indicate s.d.?

Fig1F: Statistical analysis should be performed.

Two-sample t-tests such as Fig 2J, 3H, etc:

The reviewer thinks Welch's t-test with two-tailed p-value is more suitable than Student t-test that assumes equal variance. Nonparametric test may also be suitable because of the small sample size.

Fig1D

The images of S100b and Sox2 are difficult to see their expression pattern in the merged images. Please add their separate images to the magnified views.

Page23, primers,

There is no information on GAPDH primers.

Supplemental Fig. 2d:

There are some ET-1(+) cells in the ET-1 cKO brain section. Are these cells are CD31(+) endothelial cells? Are there any differences (proliferation etc.) between the progenitors that are neighboring to the ET-1(+) cells and the other progenitors? Such data may provide information on the function of ET1 signaling from the endothelial cells.

Page5, Figure2H and page6, Figure 3F: Please add the mice strain used to these panels.

Page6, The explanation of Gsx2 and Sp8 is needed in the result section, which helps to understand the meaning of the Gsx2+ population and the Sp8+ population.

Figure4, D,E,H.

Examination of additional markers would provide further information to characterize the cells in the neurosphere, such as Sox2(+)Ki67(-) cells. Are these cells are neuronally differentiating cells or quiescent neural progenitor cells? Further, BrdU assays will also support the authors overall conclusions related to the proliferation.

Page7, line3: Has the specificity of BQ788 function been examined elsewhere? The authors should cite the reference(s) on this point. Does BQ788 injection in the Ednr β cKO provide no additional effect on the SVZ cells?

Figure panels are not mentioned in order. For example, in page5, Figure2F is mentioned before Figure2E. Many figures related to the experimental strategy are not cited in the text.

Fig6A: Developmental age should be shown in the figure.

Reviewer #3 (Remarks to the Author):

In their manuscript, Adams et al investigate critical aspects of postnatal stem cell niche development. They identified Endothelin-1 (ET-1) as important component regulating neural stem cell behavior in the developing postnatal subventricular zone (SVZ). The authors show that ET-1 and cognate receptors are expressed in the SVZ and that ET-1 maintains radial glial cells in an autocrine manner (presumably via Notch signaling). In loss of function paradigms the authors show that loss of ET-1 signaling leads to increased neurogenesis (i.e. neuroblasts giving rise to prospective granule cells in the olfactory bulb) at the expense of oligodendrocyte progenitor cells (OPC). The authors also show that ET-1 signaling is required for OPC proliferation in normal conditions and in response to experimental white matter demyelination. Interestingly, ET-1 is upregulated in CARASAL patients which show also increased numbers of OPCs. The authors propose that ET-1 may represent a future therapeutic target for cellular repair.

The manuscript by Adams et al is very clearly written and nicely illustrated. The mechanisms that control the development of the postnatal SVZ stem cell niche are not well understood. Thus the manuscript is quite timely although a few points should be addressed:

1. The authors show that *Edn1* and *Ednrb* mRNA levels are quite high in the corpus callosum (CC) and that both astrocyte and oligodendrocyte lineages express ET-1. While they show that ET-1 signaling is critical for OPCs an open question is to which extent ET-1 signaling regulates astrocyte proliferation and/or maturation during development?
2. The RNAseq experiments where the authors treat cultures with ET-1 (and thus induce ET-1 signaling) are very interesting. To be conclusive however, the reverse experiment (cultures from mice with loss of ET-1 signaling, in eg. OPC-cKO) would be important. The differences in gene expression should be quite revealing if the data from ET-1 inducing versus loss of function condition are analyzed next to each other. The results of such analysis may significantly substantiate the proposed model and claims about molecular pathways in Figure 6.
3. Related to the above point, RNAseq data from cultures of OPCs isolated from mice upon WM demyelination (WT and ET-1 cKO) are important to evaluate if 'developmental ET-1 signaling pathways' are indeed reactivated or if upregulation of ET-1 signaling has different/additional signaling functions upon demyelination. The results of these experiments will be important in the context that ET-1 signaling could be a potential future target for regeneration/cellular repair in neurodegenerative disease.

“Endothelin1 signaling maintains glial progenitor proliferation in the postnatal subventricular zone” by Adams et. al.

Response to Reviewers:

We thank the reviewers for their positive and thoughtful comments to our paper. The revised version of the manuscript is significantly strengthened by the new data generated in response to these critiques. Our study now provides mechanistic data for the role of ET-1 in subventricular zone (SVZ) neural stem and progenitor cell development, both during early mouse postnatal development and in the adult mouse SVZ following demyelination injury. Overall, our study shows the first characterization of ET-1 signaling in the SVZ and its functional roles in both neural stem cell lineage commitment and glial progenitor proliferation.

In response to the reviewers' comments, we have significantly extended our original investigation with a large body of experimental work. Specifically, the revised manuscript now includes the following additional analysis:

- New RNAseq data generated from SVZ neurospheres treated with or without ET-1, to identify downstream pathways of ET-1 signaling in radial glial cells. This analysis identified multiple neural stem cell genes that were upregulated by ET-1 treatment, as well as several proneural genes that were downregulated (Figure 4a-d). Furthermore, it confirmed our previous findings that the Notch pathway is activated by ET-1 signaling in radial glial cells (Figure 4d). These findings were validated by IHC analysis of neurospheres cultured in the absence or presence of ET-1 (new Supplemental Figure 4).
- New analysis of ET-1 cKO mice at several postnatal ages to confirm that ET-1 is functioning as an autocrine signal to promote radial glial cell maintenance and proliferation. This required crossing the ET-1 cKO strain (Nestin::CreERT2; ET-1 floxed/floxed) with Rosa26YFP reporter mice to perform lineage tracing studies at both P10 and P28 (Figure 3).
- In vivo validation of ET-1 regulated genes in OPCs in the early postnatal SVZ. We performed additional filtering of our original RNAseq data from purified SVZ OPCs to identify a more specific list of candidate genes (Figure 6d-f). We then performed RNAscope to detect potential changes in expression between SVZ OPCs in WT and Ednr β OPC-cKO mice. We found that loss of ET-1 signaling in SVZ OPCs upregulates *S100b* and *Ust* expression, while downregulating *Gsx1* expression (Figure 6g-h). Together, these results now provide direct evidence of genes regulated by ET-1 signaling in SVZ OPCs *in vivo*.
- Confirmation that developmental ET-1 signaling pathways are reactivated in the adult mouse SVZ following demyelination. Using RNAscope, we quantified changes in several genes – both within neural stem cells and OPCs – in the adult SVZ following focal demyelination of the subcortical white matter, in both WT and ET-1 cKO mice (Figure 7g-k). We found that the Notch pathway components *Jag1* and *Hes5* are reduced in the adult ET-1 cKO SVZ following injury, compared to WT mice (Figure 7g-i). We also detected changes in *Gsx1* and *S100b* expression in SVZ OPCs in adult ET-1 cKO mice following injury, compared to WT mice (Figure 7g, j, k).

We have included below a point-by-point response (in blue) for each reviewer's comments:

Reviewer 1:

This study by Adams et al. focuses on the impact of endothelin-1 on progenitor cells in the postnatal subventricular zone (SVZ). The authors conclude that endothelin-1 (ET-1) promotes radial glial cell maintenance and proliferation. Reduced ET-1 signaling increases neurogenesis and reduces oligodendrocyte progenitor cell (OPC) production and proliferation. They show that ET-1 is required for increased neural stem cell and OPC proliferation in the adult mouse SVZ after demyelination. To assess whether these data relate to human progenitor cell behavior, they studied Cathepsin A-related arteriopathy with strokes and leukoencephalopathy (CARASAL) patients. In at least one variant of this disorder, cathepsin protease activity is reduced, resulting in increased ET-1 levels. In those samples, there were increased SVZ OPCs, indicating that ET-1 also impacts SVZ progenitors in humans.

This laboratory has been investigating the impact of ET-1 on a number of cells following injury and in developing brain. This focus on the SVZ is a new and possibly quite important aspect of ET-1 function in the developing brain and after injury. As they note, few studies focus on the role of ET-1 in early development and SVZ specification. The data demonstrating signaling via ET-1 and its receptor at the SVZ are strong. The figures generally present strong and convincing data. The authors provide a model for endothelin signaling that impacts neuronal and oligodendrocyte specification during development and after injury.

We thank the reviewer for their positive comments on our manuscript and his/her appreciation that the role of ET-1 in the SVZ is an important aspect of its function in the developing brain. A detailed response to each point is provided below.

While these are clear experiments, some issues should be addressed. The most crucial is the significance of this work, relative to their earlier work on the impact of ET-1 signaling in oligodendrocytes. In general, the data here are consistent with those from their earlier work. It is encouraging that they see comparable outcomes from loss or overexpression of the ET-1 signaling system, but it is not unexpected. The unique and most important aspect of this work is the focus on SVZ cells and outcomes with respect to neuronal vs oligodendrocyte lineage commitment.

With respect to the latter part of the results, the overall conclusions are comparable to their earlier work. They note that their earlier work showed that ET-1 inhibits oligodendrocyte maturation and remyelination and suggest that contrasts with the current work showing that loss of ET-1 reduced OPC numbers and proliferation. This is not necessarily a contrasting observation, since ET-1 driving OPC proliferation would reduce their maturation and remyelination. If they directly compare OPC proliferation near the SVZ to that in medial or lateral corpus callosum in an injury state, does ET-1 impact OPCs differently? Seems unlikely, but is so, that would be important to demonstrate. They state that they did not see an impact of ET-1 on OPC proliferation in their earlier work, but as they note some of those data were in vitro studies and certainly in the Hammond et al Neuron, 2013 paper, there is a non-significant increase in OPCs after lysolecithin.

To address this question, we performed lysolecithin-induced demyelination of the lateral corpus callosum in adult mice that lack ET-1 expression in astrocytes (GFAPcreERT2; ET-1 floxed) (as previously performed in Hammond et al. 2014). We quantified the percentage of proliferating OPCs in the lesion at 7 days post lysolecithin-injection (the same timepoint we analyze the SVZ in our current manuscript). We found no statistically

significant difference in OPC proliferation (Olig2+ Ki67+ cells) in demyelinated lesions of ET-1 mutant mice, compared to WT controls (15.6% +/- 3.3% WT versus 9.8% +/- 0.2% ET-1 mutants; p-value = 0.6288; Two-way ANOVA). Therefore, loss of ET-1 in the SVZ impacts OPCs differently than loss of astrocyte-derived ET-1 in the injured white matter. This may be due to differences in the microenvironment of the SVZ and the lesion, likely resulting in different downstream pathways being activated in OPCs. We have included these new results in a new Supplementary Figure 9 and added discussion in the manuscript text.

As above, the unique aspect of this work is the impact of ET-1 on SVZ cell commitment, much more than the impact on OPC/oligodendrocyte behavior and emphasizing that more would enhance this paper.

We thank the reviewer for this constructive comment and helping us to improve the original submission. To address this issue, we have performed additional experiments and made extensive revision of the manuscript text. First, we performed RNAseq of control and ET-1 treated neurospheres (24 hours post treatment) to identify downstream signaling pathways activated by ET-1 in radial glial cells. We identified 1,160 differently expressed genes between control and ET-1 treated neurospheres. Interestingly, we found that multiple genes expressed by neural stem cells/radial glial cells were significantly increased following ET-1 treatment (*Nestin*, *Vcam1*, *Vimentin*, *Aldh1L1*, *Tnr*, *Fgfr2*, *Lif*). At the same time, we saw a significant decrease in genes associated with radial glial cell differentiation, including *Ascl1* and *Sox8*. Importantly, we also saw an increase in expression of *Jag1* (a ligand for the Notch pathway) and *Hey1* (Notch target gene), which is in agreement with our qPCR results. Interestingly, our RNAseq results also identified a significant decrease in *Hes6*, which has been previously shown to promote neuronal differentiation (Gratton et. al. 2003, Jhas et. al. 2006) and inhibit cell proliferation (Eun et. al. 2008). These results are now presented in a new Figure 4.

To validate the RNAseq results, we then performed immunohistochemical analysis of neurospheres that had been cultured in the presence or absence of ET-1 for 6 days. We found that ET-1 treated neurospheres expressed higher levels of radial glial cell markers Nestin and BLBP. We also detected Vcam1+ cells within the ET-1 treated neurospheres, unlike control neurospheres. Additionally, we found that ET-1 treatment reduced the number of Ascl1+ intermediate progenitor cells and Sp8+/Dcx+ neuronal progenitors. Interestingly, we saw a slight increase (p=0.05) in the number of Olig2+/PDGFRa+ OPCs following ET-1 treatment. This is probably due to conflicting roles of ET-1 within the neurospheres: inhibiting radial glial cell differentiation and promoting proliferation of OPCs. This data is presented in a new Supplementary Figure 4.

Together these results support our previous findings that ET-1 promotes maintenance of radial glial cells and that loss of ET-1 results in increased neurogenesis. The new data provides increased understanding of the downstream molecular pathways activated by ET-1 to promote radial glial cell identity and to repress differentiation. We have rewritten a substantial portion of the results and discussion sections to include these new results and their significance.

The other important element of this work is the demonstration that this response also occurs in human tissue. They have shown this in earlier studies in tissue from multiple sclerosis patients but highlighting and expanding both these elements would focus this paper to the concepts that make this study novel.

Our previous study of ET-1 signaling following subcortical white matter demyelination (Hammond et al. 2014) presented our finding that reactive astrocytes within the white matter lesions express high levels of ET-1. In our current manuscript, we now examine the adult human SVZ of both control and Cathepsin A-related arteriopathy with strokes and leukoencephalopathy (CARASAL) patients. CARASAL is a very rare type of leukodystrophy and was only identified in 2016 (Bugiani et al. Neurology 2016). Therefore, tissue samples for histological analysis are very precious. We found that CARASAL patients exhibit increased levels of ET-1 protein within the SVZ, which correlates with an increase in the number of GFAP+ astrocytes and PDGFRa+ OPCs within the SVZ. Although ET-1 has been previously reported to be expressed in multiple regions of the adult human brain (Naidoo et al. 2004), this is the first analysis of its expression within the human SVZ, to our knowledge.

We appreciate reviewer 1's comment and agree that it is important to determine whether ET-1's role on SVZ cell specification is conserved in the human SVZ. However, this is a very difficult hypothesis to test for several reasons: 1) unlike mouse, the human SVZ generates very few new neuronal progenitors in adulthood (Sorrells et al. 2018); 2) lineage tracing studies are impossible in human tissue; and 3) increased difficulty of staining human tissue and validating antibody specificity. Therefore, we feel that our current analysis of the CARASAL SVZ is sufficiently novel for publication of this manuscript. We are planning to investigate the role of ET-1 in regulating human SVZ progenitors in future work, and currently collecting more human tissue to specifically investigate this question in more detail.

Specific comments:

The loss of ET-1 or Ednrb from SVZ radial glial cells reduces their proliferation and shifts their commitment to the neuronal lineage without impacting the number of Olig2+ or S100B+ cells. It would be helpful to provide discussion of how the reduced number of radial glial cells and the increased numbers of neurons occur without any impact on oligodendrocyte numbers.

We completely agree with the reviewer that this is an interesting question, with several possible explanations. It is possible that the Nestin-CreER^{T2} mouse strain we use (or our tamoxifen protocol) selectively induces recombination in a subset of radial glial cells. We know this happens to a certain extent from analyzing the expression of the R26RYFP reporter throughout the SVZ. We see the highest number of YFP+ cells in the dorsal lateral SVZ with much less expression in ventral SVZ (Supplementary Figure 2). It is possible that we are primarily targeting radial glial cells that generate neuronal progenitors. In support of this hypothesis, the majority of YFP labeled cells in the WT animals are either BLBP+ or Dcx+, with only 7% expressing Olig2 (Figure 3c). Another possibility is that our experimental paradigm does not provide enough time for sufficient levels of NSC-derived gliogenesis to occur within the SVZ in order to detect a difference in our knockout mice. Therefore, we performed a second analysis at P14. Interestingly, we found a significant decrease in the percentage of Olig2+ YFP+ cells in the Ednrb cKO mice (8.5% +/- 1.4%), compared to WT mice (13.7% +/- 0.7%) (p-value = 0.0495; Welch's t-test). Therefore, reducing the number of radial glial cells does affect the number of OPCs by P14 and we have included this data in the text of the results section on page 6.

Additionally, do the SVZ radial glial cells continue to have reduced ET-1 or Ednrb by the time there are excess NeuN+ neurons in the olfactory bulb or do radial glial cells that escape deletion outcompete the poorly proliferating ET-1- or Ednrb-null cells.

We analyzed the SVZ of our Ednrb cKO mice at P28 and found that there was a reduction in the percentage of YFP+ (recombined) radial glial cells, compared to our WT mice (70.33% WT versus 43.83% Ednrb cKO mice). This indicates that the radial glial cells that escape deletion do outcompete the poorly proliferating null cells. We have added a sentence describing this finding to the manuscript in the results section on page 6.

Do the excess neurons in the olfactory bulb differentiate to functioning neurons or do they die in the bulb?

We analyzed the olfactory bulbs of P28 WT, ET-1 cKO, and Ednrb cKO mice for a marker of apoptosis – cleaved Caspase 3. Overall, we found very few Caspase3+ cells in the olfactory bulbs and did not see a change in the ET-1 cKO and Ednrb cKO mice. This suggests that the excess neurons in the olfactory bulb do not die. This data has been added to Figure 3 (Figure 3i) and the text of the results section on page 6. While we agree that it is interesting to determine whether these excess neurons differentiate to functioning neurons, we feel that this is beyond the scope of this paper. To correctly assess function, we would have to perform electrophysiological recordings of the YFP+ neurons in the olfactory bulbs at different developmental stages, which are time-consuming experiments. As olfactory bulb interneuron maturation is not the focus of this paper, these experiments will be performed in a subsequent study.

The authors demonstrate that following lysolecithin-induced demyelination in adults, there is normally increased proliferation of neural stem cells at the SVZ and an increase in OPCs. This has been seen in other contexts. One question is whether this is a shift from the normal production of neuronal precursors that move up the rostral migratory stream or rather an additional element of differentiation to the oligodendrocyte lineage from the increased number of SVZ precursors? Other studies show that after TBI, there are increased neuroblasts migrating through the rostral migratory stream, whereas MPTP reduces the number (Dixon et al., 2016; He and Nakayama, 2015).

We thank the reviewer for this comment, as we also believe that this is a very interesting question. Previous work by our lab and others has shown that a subset of neuronal precursors do change identity to become OPCs following demyelination (Jablonska et al. 2010). We expanded our analysis of the SVZ following lysolecithin (LPC)-induced demyelination in both WT and ET-1 cKO mice (new Figure 7). We analyzed the number of Sp8+ Dcx+ neuronal progenitors within the dorsolateral SVZ but found no significant differences between saline and LPC-injections for both WT and ET-1 cKO mice (Fig. 7f). This data would indicate that the increase in OPCs within the SVZ is due to either an increased number of SVZ progenitors differentiating to the oligodendrocyte lineage or increased proliferation of existing OPCs within the SVZ. The exact mechanism(s) underlying these changes will be further explored in our subsequent studies.

When damage to the olfactory bulb occurs, is ET-1 involved in the commitment to neuroblasts and their proliferation, or is this exclusively an impact of ET-1 on differentiation of SVZ progenitors to the oligodendrocyte lineage?

This is a very interesting and intriguing question. However, we feel that an additional injury model is out of scope for this paper, as the primary focus is on the developmental role of ET-1 in the early postnatal brain.

With respect specifically to the figures:

In Figure 1, it is very hard to distinguish individual cells stained for GFAP, S100b, BLBP, ET-1 or Ednrb. Thus, it is hard to assess how the numbers of percent of specific cell populations would be generated.

We have remade Figure 1 to include single-channel images for all the antibody stains.

Additionally, it is unclear what everything is compared to in A-C? B and C show changes over development, but as in A, the identification of what 1 represents should be defined in the figure legend.

We have rewritten the figure legend to include explanation of what everything is compared to in Figures 1a-c. For Figure 1a: the SCWM and cortex samples were compared to the SVZ samples, which were normalized to 1. For Figures 1b and 1c: the P1, P18, and P36 samples were compared to the P9 samples, which were normalized to 1.

Reviewer 2:

In this manuscript, the authors revealed the ET-1 signaling function in the early postnatal SVZ. Using cKOs of ET-1 and one of its receptor Ednrb, they showed that ET-1 promotes RGC proliferation via Notch signaling in an autocrine manner and OPC proliferation in a paracrine manner in normal development. They also showed ET-1 signaling is required for NSC and OPC proliferation in the adult SVZ after demyelination and suggested that such function is conserved in the human SVZ by examining the brain of the CARASAL patients.

This manuscript provides solid evidence regarding ET-1 functions in the postnatal SVZ. Their finding that ET-1 signaling promotes OPC proliferation in the SVZ is novel, and of note, it is interesting that loss of ET-1 signaling increases postnatal neurogenesis; however, the reviewer also thinks the overall findings of this manuscript are consistent with, and similar to the previous reports regarding the function of ET-1 signaling pathway in the pathological condition (Gadea et al., 2009, Hammond et al., 2014, 2015). It has been known that the RGCs and reactive astrocytes share similar characters; the RNAseq analysis in this manuscript regarding ET-1 signaling in OPCs further supports the similarity to the reactive gliosis. In this manuscript, the authors show that ET-1 signaling is conserved in the human SVZ in the pathological condition (CARASAL patients), but the involvement of ET-1 signaling in CARASAL has already reported in WM astrocytes (Bugiani et al., 2016), and the authors extended the examination to the SVZ glial progenitors. The authors discuss the dissimilarity between physiological SVZ and the pathological condition in the point of ET-1 function on OPC proliferation; however, the reviewer still wonders whether this manuscript would provide enough high impact on the research field.

We thank reviewer 2 for his/her response, and positive and constructive comments on our manuscript. In order to enhance the novelty of our manuscript we have now significantly revised it to emphasize our findings on the role of ET-1 in SVZ radial glial cell and progenitor development. We feel that these findings will provide a high impact on the research field as ET-1 is currently viewed as a molecule that prevents brain

regeneration in neurodegenerative disease and injury. It is important to recognize that ET-1 also plays important roles during development, which must be taken into account for targeted therapies. For more detail, please see our response above to Reviewer 1.

Other points:

1. *Figure 2J and 3C, H: Does the ET-1 cKO at the early postnatal ages also have the long term effect on the maintenance of the adult NSCs at P28, the fate of the YFP cells at P10, and neurogenesis at P28? These experiments would clarify whether the above effects are in an autocrine manner.*

a. *Long-term effect on the maintenance of the adult NSCs at P28.*

We collected ET-1 cKO tissue at P28 following early postnatal tamoxifen administration. We analyzed the SVZ and found that there is a significant decrease in the number of VCAM1+ GFAP+ NSCs in the ET-1 cKO at P28, compared to WT controls (46.02 +/- 1.1 cells in WT versus 33.35 +/- 4.48 cells in ET-1 cKO; p-value = 0.0266; one-way ANOVA). This data has been added to Figures 2h and 2i and discussed on page 5.

b. *The cell fate commitment of YFP-labeled cells at P10.*

We performed lineage tracing analysis of ET-1 cKO mice by crossing them with the Rosa26YFP reporter mouse strain. Following tamoxifen at P4, we analyzed the YFP+ recombined cells in the SVZ at P10. We found that there is a significant decrease in the percentage of YFP+ cells that express the radial glial marker BLBP in the ET-1 cKO mice, compared to WT mice (54.9% +/- 2.59% WT versus 34.21% +/- 1.4% ET-1 cKO; p-value = 0.0004; one-way ANOVA). We also found that there is a significant increase in the percentage of YFP+ cells that express the neuronal progenitor marker Dcx in the ET-1 cKO mice, compared to WT mice (31.93% +/- 2.3% WT versus 47.33% +/- 2.9% ET-1 cKO; p-value = 0.0072; one-way ANOVA). Interestingly, this increase is significantly less than the increase seen in the Ednrb cKO mice. This data has been added to Figure 3c and discussed on page 6.

c. *SVZ-derived olfactory bulb neurogenesis at P28.*

We collected olfactory bulbs from ET-1 cKO mice (with the Rosa26YFP reporter) at P28 following early postnatal tamoxifen administration. We found that there is a significant increase in the number of SVZ-derived YFP+ NeuN+ cells in the olfactory bulbs of ET-1 cKO mice, compared to WT controls (11.56 +/- WT versus 26.25 +/- ET-1 cKO; p-value = 0.0163; one-way ANOVA). This data has been added to Figure 3h and discussed on page 6. Together, the above results provide evidence that ET-1 maintains radial glial identity as an autocrine signal.

2. *Figure 7D, E: The authors should estimate ET-1 function in the adult SVZ under physiological condition by comparison of WT and ET-1 cKO in a saline experiment.*

We performed a two-way ANOVA to compare control WT and ET-1 cKO mice that received saline and lysolecithin-injections for the analysis of NSCs, OPCs, and neural progenitors (Figures 7d, e, f). Interestingly, there was no significant difference between saline-injected WT and ET-1 cKO mice when we examined the percentage of proliferating NSCs and OPCs (Figure 7d and e, respectively). This suggests that ET-1 does not regulate NSC and OPC proliferation in the adult SVZ under physiological

conditions, at least in the time window we examined (1 week). Furthermore, there was no significant change in the number of Sp8+ Dcx+ neuronal progenitors in the dorsolateral SVZ of the ET-1 cKO mice after saline injection, compared to WT mice (Figure 7f). These results suggest that ET-1 function may differ in the healthy adult SVZ from its developmental roles. However, further detailed analysis of the effect of ET-1 and ET receptor ablation on SVZ neural stem cells and progenitors at different timepoints is needed before conclusions can be made. As this is not the focus of this current manuscript, we will perform these experiments in a subsequent study.

3. *Page 3, line 19: The explanation of abbreviation "OL" is lacking.*

We have now included "oligodendrocyte (OL)" on page 3, which is the first use of this abbreviation.

4. *Page 5, line 7 and Fig 2E: No significant difference does not "indicate" that ependymal cells are not affected. The statistically correct interpretation should be used to describe the results. Do the VCAM1(-) cells in Fig 2B represent the ependymal cells? If so, their cell density may also provide information on this point.*

We have rephrased the sentence to now say: "Interestingly, we observed no significant difference in the total number of S100 β + cells lining the dorsolateral ventricle between WT and ET-1 cKO animals, suggesting that ependymal cells were not affected (Figure 2e)." The VCAM1- cells in the SVZ wholemount images represent a mix of cells at this developmental stage, therefore we cannot definitively state that they are ependymal cells.

5. *In the bar plots throughout the manuscript, please overlay the dot plots because the sample size is small. Do the error bars indicate s.d.?*

We have replaced all graphs to include dot plots to display sample size. All error bars indicate SEM, which is stated in both the figure legends and the methods section under statistics.

6. *Fig 1F: Statistical analysis should be performed.*

Figure 1F describes what cell types within the early postnatal SVZ express ET-1 and Ednr β . The original graph in Figure 1F has now been broken up into multiple graphs (Figure 1f, g, j, k, and n). We have not performed statistical analysis on this dataset because we are not asking whether the null hypothesis can be rejected – it is simply describing the overall expression pattern. If the reviewer feels strongly that statistics should be performed, then we will be happy to oblige.

7. *Two sample t-tests such as Fig2J, 3H, etc: The reviewer thinks Welch's t-test with two-tailed p-value is more suitable than Student t-test that assumes equal variance. Nonparametric test may also be suitable because of the small sample size.*

We have replaced our Student's t tests with two-tailed Welch's t tests, as the reviewer suggested, and updated all of the p-values in the figures and figure legends.

8. *Fig 1D: The images of S100 β and Sox2 are difficult to see their expression pattern in the merged images. Please add separate images to the magnified views.*

We have remade Figure 1 to include single-channel images for all the antibody stains.

9. *Page 23, primers. There is no information on GAPDH primers.*

We apologize for not including this information previously and have now added the sequences of the GAPDH primers to the methods section.

10. *Supplemental Fig. 2d: There are some ET-1(+) cells in the ET-1 cKO brain section. Are these cells CD31(+) endothelial cells? Are there any differences (proliferation, etc.) between the progenitors that are neighboring to the ET-1(+) cells and the other progenitors? Such data may provide information on the function of ET-1 signaling from the endothelial cells.*

The remaining ET-1+ cells in the ET-1 cKO SVZ are a combination of CD31+ endothelial cells and un-recombined WT radial glial cells (as the inducible knockdown does not give 100% ablation). While we agree with Reviewer 2 that it would be interesting to know if progenitors near the ET-1+ cells are more highly proliferative, we are unable to perform this analysis due to antibody restrictions. Furthermore, the active range of ET-1 signaling within the SVZ is unknown (i.e. what would be the criteria for a “neighboring” cell?). However, there is prior evidence that endothelial cells in the vasculature provide pro-proliferative and migratory factors for OPCs (Arai and Lo 2009). Therefore, we do expect that progenitors near CD31+ endothelial cells to be more highly proliferative compared to others within the SVZ. Unfortunately, we are unable to ablate ET-1 from SVZ endothelial cells so we cannot determine the function of ET-1 signaling from the endothelial cells. However, the phenotype of our Ednrb cKO mice is, for the most part, not significantly different from our ET-1 cKO phenotype. This important observation strongly suggests that the primary ET-1 signal is from the RGCs themselves, not the endothelial cells.

11. *Page 5, Figure 2H and Page 6, Figure 3F: Please add the mouse strain used to these panels.*

Figure 2H: Due to space restrictions, we had to remove the experimental strategy schematic. We now describe the experimental paradigm in the figure legend. All mouse strains are clearly labeled in the figure.

Figure 3F: We have added the mouse strains.

12. *Page 6: The explanation of Gsx2 and Sp8 is needed in the result section, which helps to understand the meaning of the Gsx2+ population and the Sp8+ population.*

We have moved the explanation to the results section (page 6).

13. *Figure 4D, E, H: Examination of additional markers would provide further information to characterize the cells in the neurosphere, such as Sox2(+)Ki67(-) cells. Are these cells neuronally differentiating cells or quiescent neural progenitor cells? Further, BrdU assays will also support the authors overall conclusions related to the proliferation.*

We thank the reviewer for this comment, and we performed additional analysis of the neurospheres that were treated with ET-1. We have now created a new supplementary figure summarizing this analysis (Supplementary Figure 4). We found that the majority of

the cells within the neurospheres were positive for markers of radial glia (BLBP and Nestin). Exogenous ET-1 increased the expression of these proteins and induced expression of VCAM1, a marker of neural stem cells, which was never seen under control conditions. The control neurospheres also contained a small number of neuronally and glial-differentiating cells, based on expression of Ascl1, Sp8, Dcx, Olig2, and PDGFRa. ET-1 treatment reduced the percentage of Ascl1+ progenitors and the percentage of Sp8+ Dcx+ neuronal progenitors. Interestingly, ET-1 did not significantly alter the percentage of Olig2+ PDGFRa+ OPCs within the neurospheres, although there was a trend towards an increase in OPCs following ET-1 treatment (p=0.05; Welch's t-test). This is likely due to its dual roles in both inhibiting radial glial differentiation and promoting OPC proliferation. Lastly, we performed a BrdU assay to assay ET-1's effect on proliferation. BrdU was added to the neurosphere cultures 20 minutes after addition of ET-1. The neurospheres were then collected 4 hours later and analyzed for the percentage of BrdU+ cells. ET-1 treatment increased the percentage of BrdU+ cells within the neurospheres, compared to untreated neurospheres. Together, this additional analysis and these results support our conclusion that ET-1 promotes radial glial proliferation and maintenance.

14. *Page 7, line 3: Has the specificity of BQ788 function been examined elsewhere? The authors should cite the references(s) on this point. Does BQ788 injection in the Ednrb cKO provide no additional effect on the SVZ cells?*

BQ788 is a selective antagonist of the Ednrb receptor and is widely used in the field for this purpose. We have now cited a reference on this point in the methods section on page 20. Because our Ednrb cKO mice already have a very significant reduction of Ednrb protein levels (Supplementary Figure 2), we feel that BQ788 injection into these mice would not provide an additional effect.

15. *Figure panels are not mentioned in order. For example, in page 5, Figure 2F is mentioned before Figure 2E. Many figures related to the experimental strategy are not cited in the text.*

We now cite all figures, including experimental strategy figures, in the text in order.

16. *Figure 6A: developmental age should be shown in the figure.*

We have added the developmental age of the mice used for OPC immunopanning to Figure 6a.

Reviewer 3:

In their manuscript, Adams et al investigate critical aspects of postnatal stem cell niche development. They identified Endothelin-1 (ET-1) as important component regulating neural stem cell behavior in the developing postnatal subventricular zone (SVZ). The authors show that ET-1 and cognate receptors are expressed in the SVZ and that ET-1 maintains radial glial cells in an autocrine manner (presumably via Notch signaling). In loss of function paradigms the authors show that loss of ET-1 signaling leads to increased neurogenesis (i.e. neuroblasts giving rise to prospective granule cells in the olfactory bulb) at the expense of oligodendrocyte progenitor cells (OPC). The authors also show that ET-1 signaling is required for OPC proliferation in normal conditions and in response to experimental white matter demyelination.

Interestingly, ET-1 is upregulated in CARASAL patients which show also increased numbers of OPCs. The authors propose that ET-1 may represent a future therapeutic target for cellular repair.

The manuscript by Adams et al is very clearly written and nicely illustrated. The mechanisms that control the development of the postnatal SVZ stem cell niche are not well understood. Thus the manuscript is quite timely although a few points should be addressed:

We thank Reviewer 3 for his/her positive comments and acknowledgement that our manuscript addresses an area of SVZ development that has been understudied.

1. *The authors show that Edn1 and Ednrb mRNA levels are quite high in the corpus callosum (CC) and that both astrocyte and oligodendrocyte lineages express ET-1. While they show that ET-1 signaling is critical for OPCs, an open question is to which extent ET-1 signaling regulates astrocyte proliferation and/or maturation during development?*

To directly address this comment, we ablated ET-1 from astrocytes using the GFAPCreERT2 mouse strain crossed to the ET-1 floxed mouse strain (as previously reported in Hammond et. al 2014). We used the same experimental protocol as the other analyses in this manuscript (tamoxifen at P4 and sacrificed at P10) and analyzed astrocytes within the subcortical white matter, focusing specifically on the corpus callosum (cc) and cingulum (cg) regions. We found no significant difference in the percentage of Sox9+ astrocytes that expressed GFAP or Aldh1L1, suggesting that ablation of ET-1 did not affect their maturation. However, there was a significant decrease in the percentage of proliferating Sox9+ astrocytes in the ET-1 cKO mutants, compared to WT controls (14.91% +/- 2% WT versus 7.89% +/- 1.78% ET-1 cKO; p-value = 0.0399; Welch's t-test). Interestingly, this was only observed in the cg region, as we did not see any difference in astrocyte proliferation in the cc. These results are presented in a new Supplementary Figure 8. Together, this suggests that ET-1 regulates astrocyte proliferation in specific regions of the subcortical white matter during development.

2. *The RNAseq experiments where the authors treat cultures with ET-1 (and thus induce ET-1 signaling) are very interesting. To be conclusive however, the reverse experiment (cultures from mice with loss of ET-1 signaling, in eg. OPC-cKO) would be important. The differences in gene expression should be quite revealing if the data from ET-1 inducing versus loss of function condition are analyzed next to each other. The results of such analysis may significantly substantiate the proposed model and claims about molecular pathways in Figure 6.*

While we agree with Reviewer 3 that RNAseq of OPCs purified from Ednrb OPC-cKO mice would be a nice complement to our ET-1 overexpression RNAseq results, we feel that this experiment is not needed for publication of this manuscript. It is very technically challenging to purify knockout OPCs from the SVZs of young postnatal Ednrb OPC-cKO mice (and controls) to generate sufficient, high-quality RNA for RNA-sequencing. Furthermore, our RNAseq of OPC cultures treated with ET-1 (Figure 6a-f) only resulted in 78 significantly differentially expressed genes (DEGs), which is a relatively low number (i.e. potentially easy to screen). To further refine this list and extend our molecular analysis, we compared these 78 DEGs to the new DEGs identified from the

RNAseq of neurospheres treated with ET-1 (now presented in Figure 4), in order to identify potential OPC-specific DEGs. This resulted in 41 DEGs (Figure 6d and 6f), of which we selected several to validate *in vivo* using RNAscope. We found that *S100b* and *Ust* (both downregulated in OPCs following ET-1 treatment) were significantly increased in OPCs within the SVZ of *Ednrb* OPC-cKO mice, compared to WT mice. We also found that *Gsx1* (upregulated in OPCs following ET-1 treatment) was significantly decreased in *Ednrb* OPC-cKO mice. Therefore, we have definitively identified three downstream genes in OPCs that are regulated by ET-1 signaling *in vivo*. We have included these new results in Figure 6g and 6h, and added text discussing the significance of these findings in the results and discussion sections.

3. *Related to the above point, RNAseq data from cultures of OPCs isolated from mice upon WM demyelination (WT and ET-1 cKO) are important to evaluate if 'developmental ET-1 signaling pathways' are indeed reactivated or if upregulation of ET-1 signaling has different/additional signaling functions upon demyelination. The results of these experiments will be important in the context that ET-1 signaling could be a potential future target for regeneration/cellular repair in neurodegenerative disease.*

We thank the reviewer for this comment and agree that RNAseq of the SVZ following WM demyelination would be a very interesting experiment to perform. However, we feel that their suggestion to culture OPCs isolated from the SVZ following WM demyelination of our WT and ET-1 cKO mice would not be the best approach for addressing this question, as the process of isolating and culturing the OPCs would likely change their transcriptional profiles. Therefore, in order to address this comment and to determine whether ET-1 signaling pathways are reactivated in the adult SVZ after WM demyelination, we analyzed the expression of multiple genes that we identified in the manuscript as being downstream of ET-1 signaling in the developing postnatal SVZ. To do this, we performed additional WM demyelination injuries on control and knockout mice and collected new tissue that was specifically processed for RNAscope. We then analyzed the mRNA expression of *Jag1*, *Hes5*, *Gsx1*, and *S100b* within the dorsal lateral SVZ of both WT and ET-1 cKO mice at 7 days post LPC injection. We found a significant decrease in both *Jag1* and *Hes5* in our ET-1 cKO mice, suggesting that ET-1 also activates Notch signaling in the adult SVZ after demyelination. We also found a significant decrease in the percentage of OPCs (identified by *Olig2* expression) that expressed *Gsx1* in the ET-1 cKO mice, compared to WT mice (29.17% +/- 4.11% WT versus 2.56% +/- 4.44% ET-1 cKO; p-value = 0.0003; two-way ANOVA). Lastly, we found a significant increase in the percentage of OPCs that expressed *S100b* in the ET-1 cKO mice, compared to WT mice (29.63% +/- 3.66% WT versus 65% +/- 3.76% ET-1 cKO; p-value = 0.0187; two-way ANOVA). These results are now presented in Figure 7 (7g-k) and discussed in the results and discussion sections of the text. Together, these results recapitulate our previous findings of signaling pathways downstream of ET-1 signaling in the early postnatal SVZ, indicating that they are reactivated in the adult SVZ after demyelination.

REVIEWERS' COMMENTS:

Reviewer #1 (Remarks to the Author):

This is a resubmission of a manuscript from Adams et al., focused on the impact of endothelin -1 signaling in the postnatal subventricular zone (SVZ). The authors have extensively revised the manuscript, adding data and reworking aspects of the paper. The most important element of the earlier critiques was that the uniqueness of this paper focused on the role of endothelin in the SVZ was not obvious. The authors did extensive new experiments to address that and the other concerns and they have clearly focused this paper on the SVZ. They establish that endothelin signaling in the SVZ impacts neurogenesis vs oligodendrogenesis. This is likely the first demonstration of a signal that regulates that commitment step in the SVZ. They note some important differences for endothelin signaling in oligodendrocytes in the parenchyma and SVZ, which is useful. Overall this is a significantly improved manuscript.

Reviewer #2 (Remarks to the Author):

The authors completed much additional work that greatly enhances the quality of the study. They also significantly revised the text in the results and discussion part, which would help the readers to understand the novelty and significant points of their work.

The authors have addressed all my concerns regarding the figures and experimental details in the previous manuscript. In this manuscript, I have one minor concern--- new Fig4c shows Fgfr2 and Sox8 data, but these genes are not mentioned in the text. Both of them seem to be the RGC/NSC gene; however, Sox8 is downregulated by ET1. Providing an interpretation or discussion on this point may help understand the results.

Reviewer #3 (Remarks to the Author):

The authors have addressed the points raised in the initial review quite well. They also added a lot of new data in response to all reviewers feedback that strengthen the manuscript. While I still think that RNAseq from conditions with loss of ET-1 signaling (point 2 in initial review) would be valuable and important I also tend to agree with the authors that this could be done in a future study. The presented data in the current revised version of the manuscript are strong in any case.

Response to reviewers:

The authors greatly appreciate the careful and constructive comments that all referees have provided throughout the course of this peer review process. We are happy to have adequately addressed all referee concerns and look forward to the revised manuscript being accepted and published.

Reviewer #1

(Remarks to the Author):

This is a resubmission of a manuscript from Adams et al., focused on the impact of endothelin -1 signaling in the postnatal subventricular zone (SVZ). The authors have extensively revised the manuscript, adding data and reworking aspects of the paper. The most important element of the earlier critiques was that the uniqueness of this paper focused on the role of endothelin in the SVZ was not obvious. The authors did extensive new experiments to address that and the other concerns and they have clearly focused this paper on the SVZ. They establish that endothelin signaling in the SVZ impacts neurogenesis vs oligodendrogenesis. This is likely the first demonstration of a signal that regulates that commitment step in the SVZ. They note some important differences for endothelin signaling in oligodendrocytes in the parenchyma and SVZ, which is useful. Overall this is a significantly improved manuscript.

We thank reviewer 1 for his/her comments and helpful criticism during the peer review process.

Reviewer #2

(Remarks to the Author):

The authors completed much additional work that greatly enhances the quality of the study. They also significantly revised the text in the results and discussion part, which would help the readers to understand the novelty and significant points of their work.

The authors have addressed all my concerns regarding the figures and experimental details in the previous manuscript. In this manuscript, I have one minor concern--- new Fig4c shows Fgfr2 and Sox8 data, but these genes are not mentioned in the text. Both of them seem to be the RGC/NSC gene; however, Sox8 is downregulated by ET1. Providing an interpretation or discussion on this point may help understand the results.

We thank reviewer 2 for his/her remarks and insights during the peer review process. We have addressed the concern regarding Figure 4c as follows: Fgfr2 is normally expressed by NSCs and we have added it to the list of genes in our "stem cell" network that ET-1 upregulates. Sox8 is not expressed by RGCs/NSCs and has been shown to regulate OL development in a similar fashion as Sox9 and Sox10 – promoting both OL generation from NSCs and terminal differentiation of OLs. Therefore, the fact that ET-1 downregulates Sox8 expression in neurospheres supports our conclusion that ET-1 prevents RGC differentiation. We have added a sentence to the neurosphere RNAseq results section on page 7 stating Sox8's function in OL differentiation.

Reviewer #3

(Remarks to the Author):

The authors have addressed the points raised in the initial review quite well. They also added a lot of new data in response to all reviewers feedback that strengthen the manuscript. While I still think that RNAseq from conditions with loss of ET-1 signaling (point 2 in initial review) would be valuable and important I also tend to agree with the authors that this could be done in a future study. The presented data in the current revised version of the manuscript are strong in any case.

We thank reviewer 3 for his/her positive comments and approval of our revised manuscript. We appreciate reviewer 3's willingness to defer RNAseq of ET-1 knockout OPCs to a future study.